

# Timing of land-ocean groundwater nutrient fluxes from a tropical karstic region (southern Java, Indonesia)

[1]Oehler, Till; [2]Eiche, Elisabeth; [3]Putra, Doni; [1]Adyasari, Dini; [4]Hennig, Hanna; [4]Mallast, Ulf; [1]Moosdorf, Nils

5  [1]Leibniz Centre for Tropical Marine Research (ZMT), Fahrenheitstraße 6, 28359 Bremen, Germany
[2]Karlsruher Institut für Technologie (KIT), Adenauerring 20b, 76131 Karlsruhe, Germany
[3]Universitas Gadjah Mada (UGM), Jl. Grafika 2, 55281, Yogyakarta, Indonesia
[4]Helmholtz Centre for Environmental Research GmbH (UFZ), Theodor-Lieser-Strasse 4, 06120 Halle, Germany

*Correspondence to*: Till Oehler (Till.Oehler@leibniz-zmt.de)

10  **Abstract.**

In tropical karstic regions, knowledge about the timing and quantity of land-ocean groundwater nutrient fluxes is important, as those nutrients may affect coastal ecosystems and contaminate coastal springs. High aquifer permeability of the karst, combined with high recharge and discharge during heavy rain events, leads to a close connectivity between groundwater in the hinterland and the coastal zone. The alteration between drier periods and heavy rain events can lead to a high temporal variability of groundwater discharge associated nutrient fluxes from the hinterland towards the coastal. We studied the timing of land-ocean groundwater nutrient fluxes in the tropical karstic region of Gunung Kidul (southern Java Indonesia) from November 2015 until December 2016. Satellite infrared imagery revealed two major areas of direct submarine and coastal groundwater discharge. $\delta^{18}O$ and $\delta D$ signatures, nutrient concentrations, combined with precipitation and groundwater discharge data, indicate a rapid groundwater recharge and transport from the catchment area towards the coastal ocean. Measured groundwater discharge rates varied from less than 1 m³/s up to 16.6 m³/s and were dominantly controlled by recharge in the hinterland and surface infiltration during the rainy season. Nitrate fluxes ranged from $5 \times 10^3$ to $139 \times 10^3$ mol/day and DSi fluxes from $50 \times 10^3$ to $310 \times 10^3$ mol/day. High nitrate concentrations coinciding with phases of high discharge lead to particularly high nitrate fluxes. This counterintuitive temporal connection might be due to fertilization during the onset of the wet season and the retention of nutrients from untreated sewage in the soil and in sinkholes during dryer periods, which are then washed into the aquifer during heavy rain events. In the tropical karstic region of southern Java, extraordinarily high land-ocean nutrient fluxes occur therefore during the onset of periods with high discharge, which makes coastal water and coastal springs prone to contamination during this time, while flood recession and dry periods are characterized by lower nutrient fluxes. In tropical karstic regions the timing of land-ocean groundwater nutrient fluxes is thus highly variable, which may lead to ecological implications. High nutrient fluxes during certain times of the year may explain the sudden occurrence of harmful algae blooms in coastal environments and have to be considered in coastal groundwater management.

## 1 Introduction

Groundwater discharge into the coastal ocean occurs along the worlds coastlines at the land-ocean interface and has
35  been identified as an important source of nutrients to many coastal ecosystems (e.g. Slomp and Cappellen 2004;



Paytan et al. 2006). Nutrients delivered to coastal ecosystem via groundwater discharge can lead to a shift in phytoplankton community structure, degradation of health of coral reefs and seagrass beds as reported in a number of locations all over the world (D'elia et al. 1981; Lapointe 1997; Paytan et al. 2006; Haynes et al. 2007; Chérubin et al. 2008). In tropical karstic regions, groundwater nutrient fluxes may be high, due to a high aquifer permeability and

high recharge in the hinterland during heavy rain events, high infiltration rates and low retention time of nutrients in the aquifer. Nutrient inputs into groundwater for example from anthropogenic sources in the hinterland such as fertilizers or sewage may thus rapidly reach the coastal ocean. These conditions may further lead to a high temporal variability of land-ocean groundwater nutrient fluxes.

Several studies describe the transport of nutrients towards the ocean in tropical karstic regions as in northwest

Yucatan (Mexico) (Hanshaw and Back 1980; Herrera-Silveira 1998; Young et al. 2008; Null et al. 2014), Bermuda (Lapointe and O'Connell 1989; Simmons and Lyons 1994), Barbados (Lewis 1987) and Guam (Redding et al. 2013). Yet, none focuses on the temporal behaviour of land-ocean groundwater nutrient fluxes despite of the extraordinarily importance in this setting.

Hence, at the example of the tropical karstic coastal area of Gunung Kidul (southern Java, Indonesia) this study will

i) identify groundwater recharge and flow from the catchment area to the coastal zone, and ii) elucidate the temporal behavior of groundwater nutrient fluxes in relation to precipitation backed by long-term nutrient monitoring.

## 2 Study site

### 2.1 Climate, geology and land use

The karstic region of Gunung Kidul is located in southern central Java (Indonesia). The area has a warm and moist

tropical climate with a mean annual temperature of 27 °C, a high humidity of ~80%, and an annual precipitation of up to 2000 mm (Flathe and Pfeiffer 1965; Haryono and Day 2004; Brunsch et al. 2011). The amount of precipitation is controlled by the Australian-Indonesian Summer Monsoon (Brunsch et al. 2011). The wet season lasts from November until April with precipitation rates of 150-350 mm per month. During dry season (May until October) precipitation rates are much lower with 25 to 150 mm per month (Brunsch et al. 2011). The El Niño–southern

oscillation (ENSO) has a considerable impact on the amount of annual rainfall (Aldrian and Dwi Susanto 2003). While during El Niño years, the wet season starts later in the year and during the dry season precipitation is below average (Aldrian and Dwi Susanto 2003), during La Niña the rainy season lasts longer with larger amounts of rainfall (Brunsch et al. 2011).

The study area can be classified into three different geological sections. The north and north-east section comprise

mountain ranges (Figure 1) which mainly consist of sediments and volcanic deposits of Eocene and Miocene age. Further to the south the Miocene Wonosari formations represent the second section, which mainly consists of bedded lagoonal limestones (Sir MacDonald and and Partners 1984). The third section, the Gunung Sewu area, is morphologically known for its mature Kegelkarst hills. Common to all is the southward dipping towards the Indian Ocean. At the coast, cliffs with heights of 25 to 100 m are composed of strongly karstified massive coral reef-

limestone, with intercalated clay and volcanic ash lenses (van Bemmelen 1949; Flathe and Pfeiffer 1965; Waltham et al. 1983; Haryono and Day 2004).



The combination of low rainfall amounts during dry season and quick infiltration due to strong karstification leads to severe water scarcity mainly in the southern Kegelkarst area. Consequently, more than 250,000 people depend on onshore and coastal karstic springs, rain water cisterns, water trucks or subsurface rivers as water resource (Sir MacDonald and and Partners 1984; Matthies et al. 2016). This is one reason why the region is considered to be one of the poorest regions in Indonesia with a relatively low population density of 388 inhabitants per km² in the coastal area (Dittmann et al. 2011). An underground full dam was built 100 m below ground in the karst river at Bribin Sindon in 2010 (Oberle et al. 2016), which distributes water to 75.000 people in the surrounding villages (Oberle et al. 2016).

In addition to droughts, the Kegelkarst geomorphology severely reduces the usable land surface. However, more than 90% of the population depends on agriculture often with less than 0.3 ha land per family. Dry farming with soy, corn, peanuts and cassava is the dominant type of land use. Some families additionally have cattle to work on the fields and as source for manure. Artificial fertilizer is mainly applied in addition to cattle manure to balance the nutrient deficiency of the soils. Waste water in Gunung Kidul is partly discharged directly into the subsurface or collected in unsealed septic tanks (Fach and Fuchs 2010; Nayono et al. 2010). Additionally, solid waste enters into the underground river system via sinkholes (Nayono 2014).

### 2.2 Subsurface Hydrology

Perennial rivers are absent in the coastal area and scarce in the whole Gunung Kidul due to high karst permeability. Considerable surface runoff only takes place after major rain events. In the underground, however, a complex network of caves and conduits has developed due to karstification. Consequently, subsurface discharge dominates the area. A coastal freshwater spring at Pantai Baron is connected with the Wonosari-Bribin-Baron aquifer system in the hinterland (Sir MacDonald and and Partners 1984). This aquifer system is fed by different small rivers from the volcanic Panggung Masif in the north, which enter the subsurface via a perennial river at Luweng Jomblagan (Figure 1). Pantai Baron and the 25 km upstream located subsurface river dam Bribin Sindon are connected as verified by a tracer test (Sir MacDonald and and Partners 1984) with a travel time of 14 days during dry season. An increase of groundwater travel times in Gunung Kidul by a factor of 4 during the wet season when compared to the dry season was determined during a tracer test (Eiche et al. 2012) and a groundwater travel time from Bribin Sindon to Pantai Baron within less than four days can thus be expected during the wet season.

Discharge rates are measured continuously every 10 minutes at the subsurface river dam Bribin Sindon since 2010 and show variations between <1 m³/s in the dry and up to 12 m³/s in the wet season (Oberle et al. 2016). A second branch that feeds the freshwater spring at Pantai Baron mainly originates from the Wonosari Plateau and enters the subsurface at the sinking stream Kali Suci (Figure 1). Additionally, more than 2 m³/s can be expected to be fed to Pantai Baron from the Wonosari-Baron branch in rainy season.

Groundwater in Gunung Kidul was classified as Ca-HCO$_3$ dominated, with varying hydrochemistry between wet and dry season (Eiche et al. 2016). During the dry season and flood recession periods diffuse matrix-flow conditions occur, which is indicated by a higher mineralization of groundwater. During the rainy season heavy rainfall leads to point recharge from surface water through fast conduit flow, which is accompanied by an increase in turbidity and a deterioration of the water quality (Eiche et al. 2016).





## 3 Material and Methods

### 3.1 Investigation of groundwater recharge and flow towards the ocean

In order to investigate groundwater recharge and flow towards the coastal ocean multiple methods were compared including precipitation data from four climate stations, discharge measured at the subsurface river dam Bribin

Sindon, stable isotopes ($\delta^{18}$O, $\delta$D) of groundwater, and remote sensing.

Rainfall data was obtained from four climate stations (Figure 1), operated by the Indonesian governmental agency BMKG on a daily interval from October 2015 until December 2016. Stations were chosen in a way that they most likely reflect the recharge area of the Wonosari-Bribin-Baron hydrogeological system. The first station (Nglipar, Figure 1) was located in the northern part of the Wonosari plateau at an elevation of 190 m above sea level. The

second climate station Ponjong was located at an elevation of 242 m above sea level next to the subsurface river Gunung Kendil (Figure 1). Station Semanu was located close to the subsurface river Kali Suci at an elevation of 198 m above sea level, while station Tepus was located in the district of Gunung Sewu closer to the coast at an elevation of 198 m above sea level.

Discharge was monitored at the subsurface river dam Bribin Sindon on a 10 minute interval from October 2015 until

December 2016. Discharge was monitored in a way that a defined flume was present and the water level behind the dam was measured.

Discharge areas into the coastal ocean were identified using a multi-temporal satellite-based thermal infrared approach and validated using in-situ offshore electrical conductivity (EC) measurements. The multi-temporal satellite-based thermal infrared approach exploits thermal radiance information of the sea-surface given in five

Landsat TIRS scenes (WRS path/row 120/065). All applied scenes reflect the local wet season and represent low-tide status in the years 2013/2014. The pre-processing chain besides the automatic and tide-controlled scene selection is described in Mallast et al. (2014).

Given the assumption of sea-surface temperature (SST) stabilization over time (small temporal variability) at groundwater discharge sites against a high temporal SST variability of groundwater uninfluenced areas (Siebert et al.

2014), we perform a variance analysis to detect potential groundwater discharge areas. It includes calculating the median of the absolute differences per pixel between each SST scene. This reduces influences of potential outliers evoked through e.g. thin clouds and represents an advancement described in Mallast et al. (2014). The final result is a SST variance image in which small variance values represent potential groundwater discharge areas (Figure 3). These potential groundwater discharge areas were validated with in-situ offshore electrical conductivity (EC) and

temperature measurements. Measurements were taken on the 14[th], 16[th] and 19[th] of November 2015 and on the 18[th], 19[th] and 20[th] of April 2016 at a depth of 10 cm below water level with a WTW[TM] TetraCon 925-P conductivity probe from a small fishing boat.

### 3.2 Hydrochemical sampling

Hydrochemical samples were obtained from three subsurface rivers (Gua Pindul, Kali Suci, Gunung Kendil) in the

hinterland and from two coastal springs (Pantai Baron, Pantai Ngrumput) (Figure 1) which were identified based on the remote sensing approach (Figure 3). At most sites samples were taken in November 2015 and on a nearly





monthly interval from April 2016 until December 2016. Electrical conductivity, temperature, pH-values, and dissolved oxygen were measured directly in the field using handheld calibrated probes: conductivity measuring cell (WTW$^{TM}$ TetraCon 925-P), pH (WTW$^{TM}$ Sentix 940), dissolved oxygen (WTW$^{TM}$ FDO 925). Water samples were directly filtered using Whatman filters (CA 0.45 µm). Samples for stable water isotopes ($\delta^{18}$O, $\delta$D) were directly

transferred into gas tight glass vials (2 ml) and stored without headspace. In Germany, water isotopy was measured using a Picarro$^{TM}$ L2130-*i* Analyzer. Values are reported as Vienna Standard Mean Ocean Water (VSMOW).

In November 2015, April 2016, May 2016, June 2016, July 2016 and December 2016 water samples for nutrient analyses (nitrate, nitrite, ammonium, phosphate, DSi) were filtered and filled into 20 ml HDPE bottles and preserved either by freezing or by adding 50 µl of saturated HgCl$_2$ solution. These water samples were transported to Germany

and measured at the laboratory of the Leibniz Centre for Marine Tropical Research in Bremen using standard photometrical methods (Grasshoff et al. 2009). In August 2016, September 2016, October 2016 and November 2016 nitrate was determined in freshwater samples within a couple of days after sampling at the Geochemistry Laboratory, Dept. of Geological Engineering, Faculty of Engineering, UGM, Yogyakarta (Indonesia) using an ion chromatograph (MetroOhm IC-850) with an accuracy of 0.01 µM.

Nutrient concentrations and stable isotopes of water at the brackish water spring Pantai Ngrumput were normalized to freshwater according to Hunt and Rosa (2009), in order to be able to compare freshwater normalized values with the freshwater spring Pantai Baron and subsurface rivers in the hinterland.

### 3.3 Calculation of groundwater nutrient fluxes

Most of the groundwater which discharges at the subsurface river dam Bribin Sindon reaches Pantai Baron in 14

days during dry season (Sir MacDonald and and Partners 1984) and within 4 days during wet season, as estimated from a tracer test (Eiche et al. 2012). Therefore we were able to calculate a groundwater nutrient flux by multiplying the nutrient concentration at Pantai Baron with the discharge rates of the upstream located subsurface river dam Bribin Sindon (Table 2, Figure 6). During the wet season an average discharge rate from the 4$^{th}$ day prior to sampling at Pantai Baron was used, as we assumed a travel time of 4 days from Bribin Sindon to Pantai Baron. During dry

season an average discharge rate from the 14$^{th}$ day prior to sampling at Pantai Baron was used, because of the estimated travel time of 14 days from Bribin Sindon to Pantai Baron based on the tracer test by McDonald (1984).

## 4 Results

### 4.1 Groundwater recharge, flow, and discharge to the coastal ocean

Precipitation data from four climate stations, discharge data from the subsurface river dam Bribin Sindon, remote

sensing data and stable isotopes of water were compared in order to investigate groundwater recharge and flow towards the coastal ocean. Groundwater recharge in Gunung Kidul occurs either from subsurface rivers which enter the area in the north or from direct infiltration of rainwater from above. Climate stations Nglipar and Ponjong are located in the hinterland (Figure 1). Precipitation measured at these stations feeds the subsurface rivers in the north and north east and therefore reflects groundwater recharge from subsurface rivers in the hinterland. Climate stations





Semanu and Tepus are located closer to the coast and precipitation data from these stations reflects groundwater recharge from direct infiltration from above.

Precipitation data from climate stations Nglipar and Ponjong are shown together with discharge rates measured at Bribin Sindon in Figure 2. A seasonal trend of rainfall and discharge data influenced by the ENSO in the year
2015/2016 was observed. Typical for an El Niño year the rainy season started delayed in 2015 with no rainfall in October 2015, low precipitation rates in November 2015, followed by an onset of the rainy season in December 2015 which lasted until April 2016. Driest months were July 2016 and August 2016. As expected for a La Niña year, rainy season already started earlier in 2016 at the end of September 2016. Precipitation and discharge rates also varied on a daily basis. Especially during the rainy season, large amounts of daily rainfall were followed by high discharge rates
at Bribin Sindon. For example on the 11[th] of December 2015 high precipitation rates of 187 mm/day were measured at Nglipar, which was followed by an increase in discharge rates from 1.4 m³/s to 9.8 m³/s on the 16[th] of December 2015 at Bribin Sindon. In general, higher precipitation rates from mid of January until end of February lead to an increase in discharge rates at Bribin Sindon, in a similar way as during mid of April 2016. A recession period in precipitation and discharge rates was observed from end of April 2016 until the end of July 2016 (Figure 2). On the
26[th] of July, unusually high precipitation rates of 88 mm/day were observed at Nglipar which was followed by a sudden increase in discharge rates from 1.3 m³/s to 8.1 m³/s on the 1[st] of August 2016. From the 28[th] of November 2016 discharge rates increased from 3.0 m³/s to 8.0 m³/s and then further increased by the beginning of December up to 16.1 m³/s. During this time a constant high precipitation was observed at all climate stations Nglipar, Ponjong, Semanu and Tepus (Figure 2, electronic supplement).
Two sites at which groundwater discharge into the ocean may occur were identified based on low SST variability values of <0.24 (blue colors in Figure 3) on the multi-temporal satellite-based thermal infrared approach. The first site occurs within a bay-like structure to which it is bounded. It corresponds to a karstic freshwater spring, Pantai Baron, which shows electrical conductivities of 0.3-0.5 mS/cm in the nearshore area and a clear freshwater contribution due to lower EC values at a distance of up to 500 m away from the coast. The second site matches the
location of an intertidal spring located at Pantai Ngrumput (Figure 3). The spring was accessible during low tide and showed brackish water discharge along the beach in the intertidal area with EC values which varied over the seasonal cycle with around 5.95 to 6.30 mS/cm in the wet season and 7.51 to 9.53 mS/cm during the dry season.

The stable isotopes $\delta^{18}O$ and $\delta D$ of coastal springs and subsurface rivers are shown in Table 1 and Figure 4. $\delta^{18}O$ and $\delta D$ values of the subsurface rivers, and coastal springs showed a slight deviation towards lighter $\delta^{18}O$ values when
compared the global meteoric water line and a local meteoric water line (Sidauruk 2015). Especially during the wet season lighter $\delta^{18}O$ (< -7.11 ‰) and lighter $\delta D$ (< -44.25 ‰) values were observed at Pantai Baron, when compared to groundwater samples obtained in the hinterland (Table 1).

## 4.2 Temporal variability of nutrient concentrations and fluxes

Nitrate and dissolved silicon (DSi) concentrations which were measured at the subsurface rivers in the hinterland and
the coastal springs are shown in Table 1 and Figure 5A+B. DSi concentrations of subsurface river water in the hinterland and coastal groundwater showed a similar temporal variability with concentrations that ranged from about 300 µM up to 500 µM (Figure 5A). Nitrate showed a high temporal variability in both waters sampled in the



hinterland and at coastal springs with concentrations that ranged from 9 µM to 300 µM (Figure 5B, Table 1). High nitrate concentrations were observed during the rainy season in November 2015 and April 2016, while concentrations decreased at the transition from rainy season in 2016 towards the dry season from April to May 2016. Highest nitrate concentrations of up to 300 µM were observed during the dry season in August 2016.

Highest nutrient fluxes occurred in April 2016, August 2016 and in December 2016 (Table 2, Figure 6). DSi fluxes ranged from $265 \times 10^3$ mol per day during the wet season (December 2016) down to $50 \times 10^3$ mol per day during dry season (end of June). Highest nitrate fluxes of $90 \times 10^3$ mol per day (April 2016) and $139 \times 10^3$ mol per day (December 2016), were caused by a high discharge combined with high groundwater nitrate concentrations during the wet season. Nitrate fluxes and concentrations were in general high following times which were characterized by

low discharge. For example, until mid of October 2015 a low discharge of about 1 m³/s was observed, followed by higher discharge of about 3 to 4 m³/s and high nitrate concentrations of 114.5 µM in November 2015. Similarly, in August 2016 high nitrate concentrations (300 µM) and fluxes ($85 \times 10^3$ mol per day) followed a month of low discharge (about 1.3 m³/s at the end of July 2016). In December 2016 high nitrate fluxes of $139 \times 10^3$ mol per day followed an event of high discharge, while November 2016 was characterized by lower nitrate fluxes of $12 \times 10^3$ mol

per day (Figure 6).

## 5 Discussion

### 5.1 Groundwater recharge and flow towards the coastal ocean

We investigated groundwater recharge, flow and discharge into the coastal ocean using multiple methods including rainfall data, discharge data, hydrochemical data, and remote sensing. In the investigation area, a major amount of

groundwater which enters the ocean via subsurface rivers recharges in a larger distance from the coast (e.g. through sinking streams or sinkholes), as indicated by discharge measured at Bribin Sindon which lagged heavy rain events in the hinterland by about four days (Figure 2). A further part of groundwater recharge results from direct infiltration of rainwater from the surface in the hinterland (e.g. via close by sinkholes) especially during rainy season, indicated by concurrent high discharge rates and high precipitation rates measured at all climate stations in December 2016. A

general deviation towards lighter $\delta^{18}O$ values when compared to the local and the global meteoric water line (Figure 4) (Clark and Fritz 1997) indicates that no evaporative loss occurred which hints towards a rapid infiltration of rainwater into groundwater. A rapid recharge of groundwater in Gunung Kidul, especially with quick infiltration during the rainy season, with point recharge during heavy rain events was also pointed out by Eiche et al. (2016). As a consequence, coastal groundwater is prone to contamination especially during the rainy season due to rapid

infiltration, and a rapid transport of groundwater towards the coast. For example during the rainy season, values of coliform bacteria and *E. coli* in groundwater increase dramatically with extreme peaks (total coliforms >39000 CFU/100ml, *E.Coli* >9600 CFU/100ml) at the beginning of the rainy season in Gunung Kidul (Matthies et al. 2014). In Gunung Kidul, coastal freshwater springs are already used by local communities as an important freshwater source (Sir MacDonald and and Partners 1984; Moosdorf and Oehler 2017). This may even increase in the future as

the area is currently being developed for tourism. Consequently, water quality is of high concern. Since 2015 the Gunung Sewu UNESCO Global Geopark (UNESCO 2017) attracts visitors and tourists to the area. Furthermore,



hotels, resorts and a golf course are being planned in the coastal area. A deterioration of the coastal groundwater quality during wet season should thus be considered for the water management in the area.

## 5.2 Timing of land-ocean groundwater nutrient fluxes

Correlating discharge rates at karstic springs with nutrient concentrations can help identifying and characterizing the flow behavior and the source of a nutrient. A negative correlation between discharge and nutrient concentration for example indicates a constant source of nutrients which is diluted during times of high discharge. A positive correlation between discharge rates and nutrient concentrations point towards a surface source of nutrients which is enriched during dry season and washed into the aquifer as a pulse in the rainy season. In Gunung Kidul, we observed both a positive and a negative correlation between discharge rates and nutrients depending on the specific nutrient and the time when samples were taken.

A general increase of DSi concentrations during the flood recession period from April 2016 until June 2016 indicates diffuse-matrix flow conditions (Eiche et al. 2016). A lower concentration during times of higher discharge indicated a dilution effect in the aquifer during wet season during conduit flow conditions (Figure 5A). However, DSi fluxes were still higher during wet season, when compared to dry season indicating a further DSi source during wet season. Probably fine grained DSi rich volcanic sediments and soil material are eroded during times of high discharge from the Mountain ranges of Panggung in the north-east and the agricultural fields nearby. The transport colloidal material in the karstic area during the wet season was also indicated by brownish water color for example during flash floods and can often be observed in the area after heavy rain events in surface and subsurface rivers (Eiche et al. 2016).

Nitrate concentrations mostly increased during times of high discharge, especially during times when high discharge followed a dry period, such as in November 2015, April 2016, August 2016 or December 2016. Large amounts of nutrients at the start of the rain season and after events of high discharge (e.g. December 2016) may be attributed to a wash-out of nitrogen from a surface source which accumulated in the soil and in sinkholes during the dry season like fertilizers, animal waste and general village waste, which can also be observed in other karstic areas (Guo and Jiang 2009). During wet season high groundwater nutrient fluxes towards the coastal ocean were for example also observed in Florida Keys (Lapointe et al. 1990) and in Bermudas, possibly due to the use of fertilizers in the hinterland (Lewis 1987). In Gunung Kidul major crops such as rice, corn and soy are mainly fertilized during rainy season during the first two weeks of November and the first two weeks of March, when water availability is high (Katam 2017). In general three different types of fertilizer are used in the area of Gunung Kidul including Urea, SP 36, as well as potassium chloride. From these fertilizers urea contains large amounts of nitrogen in the form of ammonium, which might further be oxidized to nitrate. Furthermore large amounts of nutrients and contaminants may accumulate in the soil and in sinkholes from sewage, due to a lack of wastewater treatment in the area.

Decreasing nitrate fluxes during flood recession periods (e.g. for April to July 2016) are attributed to lower discharge rates and decreasing groundwater nitrate concentrations. Decreasing concentrations in groundwater are probably due to the temporal exhaust of the accumulated and easily soluble nitrate pool. During the next dry period, however, nitrate will again be accumulated. In karstic tropical regions the land-ocean groundwater nutrient fluxes are thus highly variable, as groundwater nutrient concentrations and groundwater discharge both show a high temporal



variability over the year and are often positively correlated. However, knowledge about groundwater recharge, flow direction and anthropogenic activity in the hinterland, may allow the prediction of times of high groundwater nutrient fluxes and thus the implementation of appropriate water management measures.

### 5.3 Uncertainties

The main uncertainties in this study concern the connectivity between the catchment area and the coastal ocean, as well as the amount of nutrients discharged into the coastal ocean. The aquifer connection between Pantai Baron and the subterranean river gauge (Bribin Sindon) was shown in several tracer tests, but unknown flow paths may exist in between, which would additionally influence discharge at Pantai Baron. A general connection between the Pantai Ngrumput and the aquifer system in the hinterland was deduced from the temporal variability of the hydrochemistry,

but no tracer test has been performed to verify this connection.

The total groundwater nutrient flux from the hinterland into the coastal area is also uncertain, because additional groundwater discharge from other small submarine and coastal springs occurs in the area. Some discharge estimates from coastal springs showed that discharge is about an order of magnitude lower at most coastal and submarine springs, when compared to Pantai Baron. For example, discharge measurements with a flow meter revealed 0.06 m³/s

at Pantai Ngrumput, and 0.2 m³/s at Pantai Sundak (Sir MacDonald and and Partners 1984) . We can still assume that the dominant amount of groundwater discharge and associated nutrient fluxes occur at Pantai Baron within the area.

### 6 Conclusion

We studied the temporal variability of nutrient fluxes into the ocean through groundwater in the tropical karstic region of Gunung Kidul, Indonesia. While in many porous coastal aquifers the transport of terrestrial groundwater

towards the ocean is slow, in the karstic region of Gunung Kidul transport of groundwater can be fast with tens of kilometers from the hinterland within a few days and also highly variable throughout the year.  Contaminants and nutrients which reach the coastal zone via groundwater discharge may thus have their source far in the hinterland. Groundwater nutrient fluxes are especially high during periods of high discharge following periods of lower discharge, due to nutrient input into the aquifer from the surface pool (fertilizers, untreated sewage, and waste) that

accumulates during dry periods. Heavy rain quickly flushes nutrients into the subsurface without considerable contaminant retention from where they are then rapidly transported towards the coastal ocean. During flood recession and dry periods, land-ocean groundwater nutrient fluxes are on the other hand mainly very low. Therefore, the timing of land ocean groundwater nutrient fluxes is highly variable in Gunung Kidul, and heavily influenced by the amount of precipitation in combination with anthropogenic activities. Similar can be expected in other tropical

karstic regions as well. Several ecological implications may arise from such peak discharge nutrient flux events, such as eutrophication of coastal waters during a certain time in the year or a sudden occurrence of harmful algae blooms, which has to be considered for the management of coastal waters and coastal groundwater. However, the temporal variability can be closely estimated if the groundwater recharge behavior, flow and anthropogenic activities in the hinterland are known. In order to capture the temporal variability a good approach might be to study fluxes during



dry conditions for a minimum flux, and after heavy rain events which followed dry periods to capture high fluxes during peak discharge events.

**Acknowledgements**

This work was funded through the BMBF junior research group SGD-NUT (Grant #01LN1307A to Nils Moosdorf). Alexandra Galisson, Rilo Restu Surya Atmaja and Adelide Asriati are gratefully acknowledged for their help with sampling. Rilo Restu Surya Atmaja is further acknowledged for acquiring rainfall data. We thank Daniel Stoffel and his colleagues from the Institute of Water and River Basin Management at KIT for providing discharge data from the subsurface river dam Bribin Sindon. Thanks also to Maren Hochschild for providing information about the tracer tests.

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




Figures

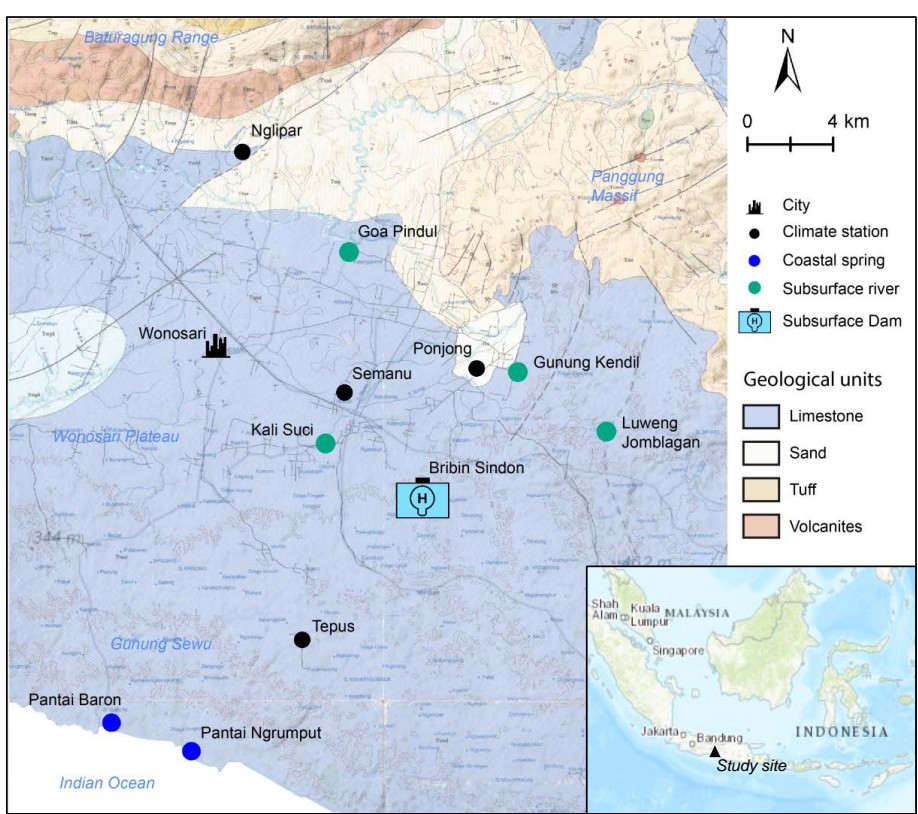

**Figure 1: A simplified geological map with the major geological units of Gunung Kidul, including the locations of the**
5    **climate stations (black), and the subsurface rivers (green), the subsurface river dam Bribin Sindon (blue square), and the coastal springs (blue). A detailed geological map can be found in** Toha and Sudarno (1992)**.**



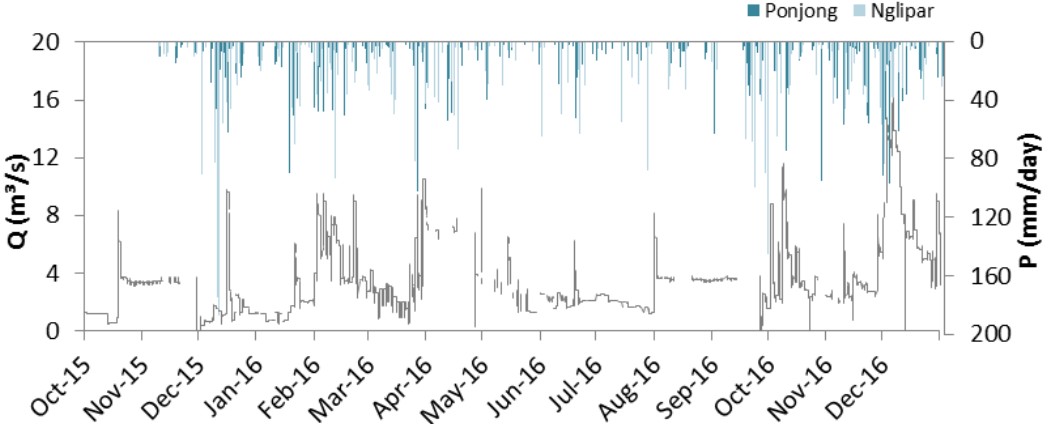

**Figure 2: Precipitation rates (upper graph) measured at the two climate stations Ponjong (dark blue) and Nglipar (bright blue) in the hinterland, and discharge rates (lower graph, grey) measured at the subsurface river dam Bribin Sidon.**

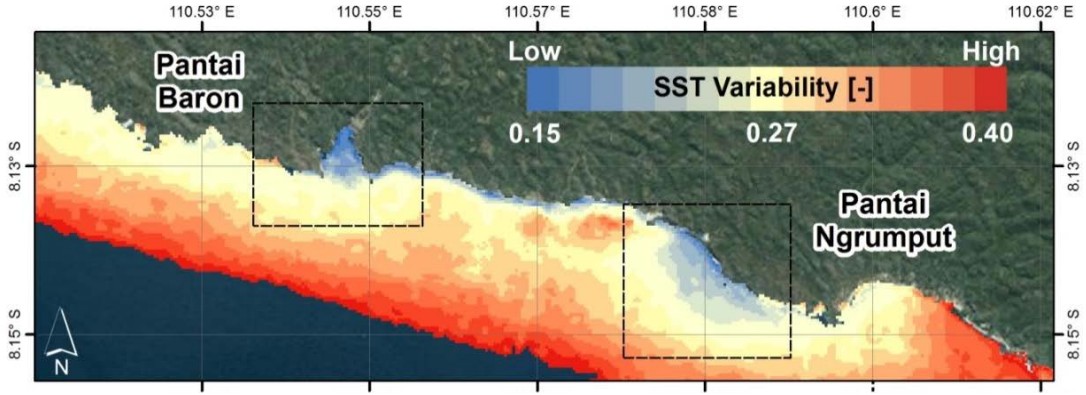

**Figure 3: The multi-temporal satellite-based thermal infrared image indicated two major sites at which groundwater discharges into the ocean.**





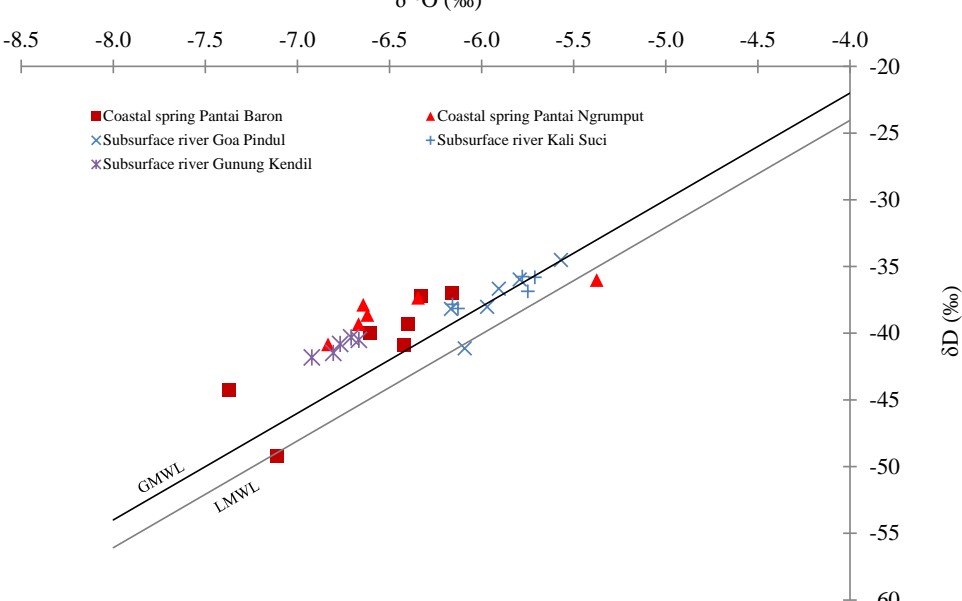

**Figure 4: Stable isotopes of coastal springs are shown in red and of subsurface rivers in the hinterland in blue and purple. A local meteoric water line is shown in grey and the global meteoric water line in black. Legend for hydrochemical samples: Subsurface rivers: Goa Pindul (blue cross), Kali Suci (blue plus), and Gunung Kendil (purple star). Coastal springs: Pantai Baron (red square) and Pantai Ngrumput (red triangle).**




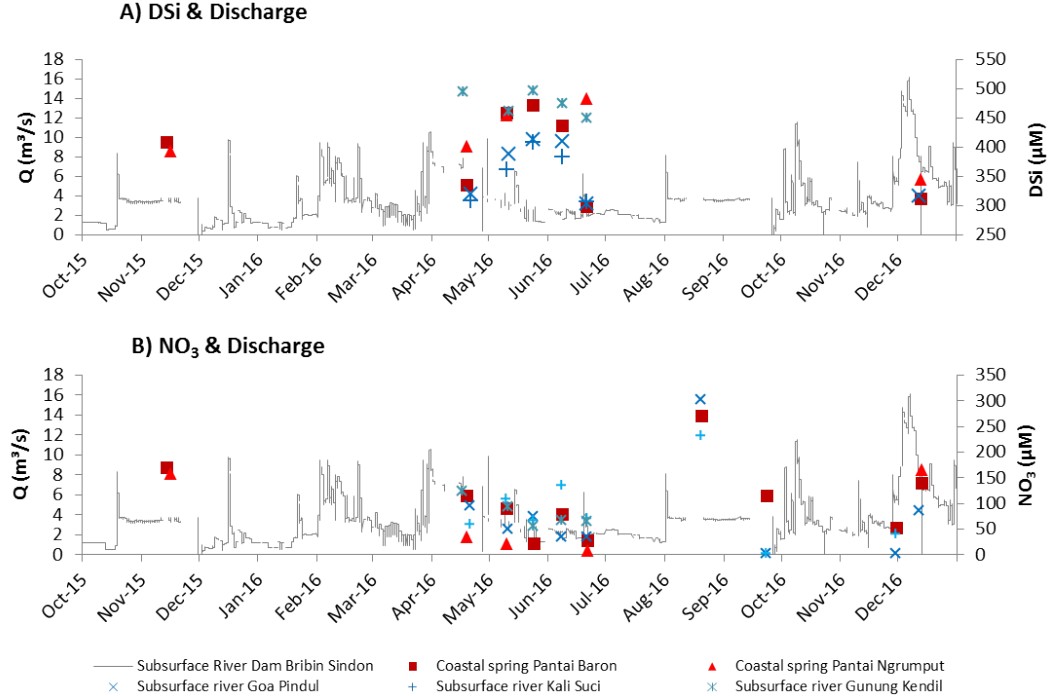

**Figure 5: Discharge rates measured upstream of the coastal spring Pantai Baron at the subsurface river dam Bribin Sindon are shown by the grey line in A) and B). DSi concentrations of subsurface rivers and coastal springs are shown in A (DSi) and B) (nitrate). Legend for hydrochemical samples: Subsurface rivers Goa Pindul (blue cross), Kali Suci (blue plus), and Gunung Kendil (purple star). Coastal springs Pantai Baron (red square), and Pantai Ngrumput (red triangle).**



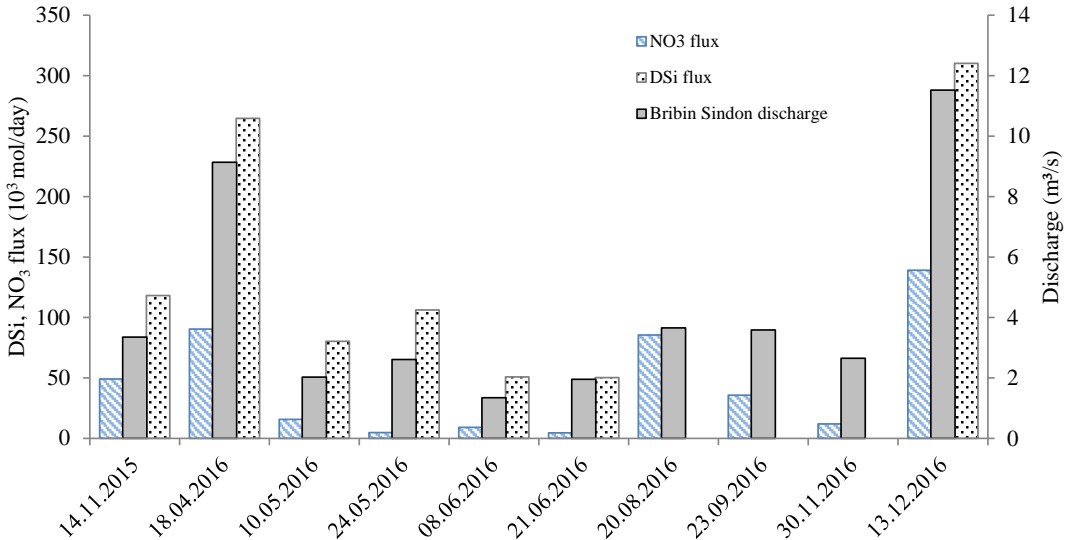

**Figure 6: Groundwater discharge (grey), as well as groundwater nitrate fluxes (blue shaded) and DSi fluxes (black dots) in 10$^3$ mol/day estimated at Pantai Baron from November 2015 until December 2016.**





4    **Tables**

5    **Table 1: Hydrochemical and physical data obtained from different sampling sites. Wet seasons are marked in grey.**
6    **BDL=below detection limit.**

| Site | Date | Season | pH | O₂ (%) | Cond. (µS/cm) | Temp (°C) | d¹⁸O (‰) | dD (‰) | NO₃ (µM) | NO₂ (µM) | DSi (µM) | NH₄ (µM) | PO₄ (µM) |
|---|---|---|---|---|---|---|---|---|---|---|---|---|---|
| Pantai Baron | 14.11.2015 | Dry | 6.97 | 83 | 557 | 27.9 | -6.60 | -40.00 | 169.8 | 0.01 | 407.9 | | 0.14 |
| | 19.04.2016 | Wet | 7.32 | 85 | 429 | 27.6 | -7.37 | -44.25 | 114.5 | 0.31 | 335.3 | 1.32 | 0.15 |
| | 10.05.2016 | Dry | 7.38 | 88 | 521 | 28.0 | -6.16 | -36.98 | 89.6 | 0.24 | 457.5 | 0.59 | 0.13 |
| | 24.05.2016 | Dry | 7.36 | 82 | 541 | 28.1 | -6.40 | -39.32 | 21.4 | 0.43 | 470.7 | 3.58 | 0.15 |
| | 08.06.2016 | Dry | 7.31 | 82 | 525 | 28.0 | -6.42 | -40.91 | 78.0 | 0.28 | 436.2 | 2.37 | 0.11 |
| | 21.06.2016 | Dry | 7.55 | 77 | 384 | 27.2 | -6.33 | -37.21 | 26.9 | 0.22 | 297.2 | 1.17 | 0.19 |
| | 20.08.2016 | Dry | | | 640 | 27.8 | | | 270.5 | | | | |
| | 23.09.2016 | Dry | | | 820 | 28.4 | | | 115.0 | | | | |
| | 30.11.2016 | Dry | | | 260 | 23.0 | | | 52.0 | | | | |
| | 13.12.2016 | Wet | | 92 | 429 | 27.6 | -7.11 | -49.21 | 139.6 | 0.42 | 311.5 | | 0.90 |
| Pantai Ngrumput | 16.11.2015 | Dry | 7.15 | 75 | 8380 | 27.8 | -6.34 | -37.35 | 158.9 | 0.02 | 393.1 | | 0.54 |
| | 19.04.2016 | Wet | 7.29 | 72 | 6300 | 28.4 | -6.67 | -39.34 | 34.6 | 0.13 | 401.8 | 7.88 | 0.14 |
| | 10.05.2016 | Dry | 7.43 | 83 | 9530 | 28.8 | -6.62 | -38.64 | 21.0 | 0.24 | 454.4 | 7.36 | 0.13 |
| | 21.06.2016 | Dry | 7.46 | 72 | 9450 | 28.4 | -5.37 | -36.02 | 9.0 | 0.40 | 483.0 | 5.06 | 0.12 |
| | 20.08.2016 | Dry | | | 7520 | 28.2 | | | | | | | |
| | 23.09.2016 | Dry | | | 7510 | 28.5 | | | | | | | |
| | 30.11.2016 | Wet | | | 8080 | 27.2 | | | | | | | |
| | 13.12.2016 | Wet | 7.5 | 67 | 5950 | 28.2 | -6.83 | -40.85 | 165.8 | 0.01 | 344.2 | | 0.97 |
| Gua Pindul | 21.04.2016 | Wet | 7.68 | 92 | 533 | 28.6 | -5.57 | -34.52 | 94.1 | 0.59 | 320.2 | 1.87 | 0.14 |
| | 11.05.2016 | Dry | 7.76 | 88 | 540 | 28.5 | -5.97 | -38.03 | 48.5 | 0.31 | 388.6 | 1.91 | 0.11 |
| | 24.05.2016 | Dry | 7.79 | 87 | 567 | 27.9 | -5.79 | -35.99 | 72.4 | 0.35 | 413.4 | 1.42 | 0.11 |
| | 08.06.2016 | Dry | 7.75 | 86 | 558 | 28.4 | -5.91 | -36.68 | 33.3 | 0.28 | 409.7 | 2.07 | 0.12 |
| | 21.06.2016 | Dry | 7.79 | 86 | 413 | 27.3 | -6.17 | -38.19 | 33.6 | 0.36 | 302.9 | 3.32 | 0.11 |
| | 20.08.2016 | Dry | | | 520 | 29.6 | | | 301.5 | | | | |
| | 23.09.2016 | Dry | | | 560 | 29.0 | | | BDL | | | | |
| | 30.11.2016 | Wet | | | 270 | 25.6 | | | BDL | | | | |
| | 12.12.2016 | Wet | | 99 | 494 | 26.1 | -6.09 | -41.14 | 83.8 | 1.13 | 315.8 | | 0.09 |
| Gunung Kendil | 17.04.2016 | Wet | 7.29 | 93 | 567 | 27.8 | -6.77 | -40.82 | 122.7 | 0.51 | 494.6 | 4.10 | 0.11 |
| | 11.05.2016 | Dry | 7.21 | 84 | 522 | 27.4 | -6.67 | -40.51 | 91.0 | 0.04 | 461.9 | 0.03 | 0.12 |
| | 24.05.2016 | Dry | 7.30 | 82 | 524 | 27.2 | -6.80 | -41.48 | 53.4 | 0.19 | 497.0 | 1.43 | 0.11 |
| | 08.06.2016 | Dry | 7.26 | 73 | 527 | 27.0 | -6.92 | -41.82 | 66.1 | 0.23 | 474.2 | 0.25 | 0.13 |
| | 21.06.2016 | Dry | 7.34 | 82 | 532 | 27.1 | -6.71 | -40.32 | 63.7 | 0.03 | 450.4 | 0.04 | 0.11 |
| Kali Suci | 21.04.2016 | Wet | 8.36 | 104 | 426 | 29.1 | | | 58.0 | 0.17 | 308.5 | 1.10 | 0.11 |
| | 10.05.2016 | Dry | 8.46 | 102 | 431 | 28.1 | -6.13 | -38.15 | 106.8 | 0.34 | 361.5 | 1.74 | 0.12 |
| | 24.05.2016 | Dry | 8.27 | 101 | 485 | 27.5 | -5.78 | -35.77 | 66.1 | 0.16 | 408.4 | 0.89 | 0.10 |
| | 08.06.2016 | Dry | 8.32 | 104 | 490 | 27.5 | -5.75 | -36.87 | 132.8 | 0.17 | 382.3 | 0.63 | 0.09 |
| | 21.06.2016 | Dry | 8.52 | 102 | 417 | 26.7 | -6.16 | -37.85 | 69.2 | 0.11 | 306.7 | 0.30 | 0.11 |
| | 20.08.2016 | Dry | | | 520 | 28.9 | | | 229.5 | | | | |
| | 23.09.2016 | Dry | | | 490 | 29.2 | | | BDL | | | | |
| | 30.11.2016 | Wet | | | 200 | 26.7 | | | 38.0 | | | | |





**Table 2: (A) Sampling dates at Pantai Baron, (B) average discharge rates measured at the subsurface river dam Bribin Sindon 4 days prior to sampling during wet season and 14 days prior to sampling during the dry season, (C) DSi Fluxes in $10^3$ mol/day, and (D) $NO_3$ fluxes in $10^3$ mol/day.**

| (A) Sampling Date | (B) Q (m³/s) | Qmin (m³/s) | Qmax (m³/s) | (C) DSi Flux ($10^3$ mol/day) | (D) $NO_3$ Flux ($10^3$ mol/day) |
|---|---|---|---|---|---|
| 14.11.2015 | 3.35 | 3.19 | 3.46 | 118 | 49 |
| 18.04.2016 | 9.14 | 6.89 | 11.63 | 265 | 90 |
| 10.05.2016 | 2.03 | 0.34 | 6.85 | 80 | 16 |
| 24.05.2016 | 2.61 | 2.54 | 3.34 | 106 | 5 |
| 08.06.2016 | 1.35 | 1.34 | 1.36 | 51 | 9 |
| 21.06.2016 | 1.96 | 1.55 | 2.34 | 50 | 5 |
| 20.08.2016 | 3.66 | 3.65 | 3.69 | NA | 85 |
| 23.09.2016 | 3.59 | 3.52 | 3.62 | NA | 36 |
| 30.11.2016 | 2.65 | 2.38 | 2.72 | NA | 12 |
| 13.12.2016 | 11.52 | 10.00 | 12.41 | 310 | 139 |

