# Peer review of "Timing of land-ocean groundwater nutrient fluxes from a tropical karstic region (southern Java, Indonesia)"

_Hydrology and Earth System Sciences, 2017_

## Referee Comment (RC1) · Anonymous Referee #1 · 27 Dec 2017

Summary:

This manuscript investigates the temporal variability of nitrate and silicate fluxes to the coastal ocean via coastal springs in Gunung Kidul, a tropical karstic region located in southern Java, Indonesia. The authors identified two major areas of groundwater discharge to the ocean using thermal infrared imaging and quantified the discharge based on continuous measurements in a gauged subsurface river dam. Multiple water samples were collected at the two coast springs during varying climatic conditions to measure nutrient concentrations and calculate groundwater-derived nutrients discharge to the ocean. The authors studied the temporal trends of groundwater nutrients

discharge during the wet season, flood recession periods, and the dry season. Their results showed that higher nitrate concentration was found during the wet season when the discharge was at its highest, posing a potential threat the coastal ecosystem due to excess nitrogen inputs. I think that this study is sufficiently relevant to the field of groundwater discharge to the ocean and should be considered for publication in HESS. However, a number of questions must be addressed before its publication.

Major comments:

I agree with the authors that the main uncertainties in this study are the connection between the catchment area and the nutrients discharge to the coastal ocean. These two points are basically the main goals of the study and a better job must be done to justify the lack of data in this concern:

1) As the authors mention, assuming that the discharge in Pantai Baron is the same as the flow measured in Bribin Sindon and can be directly derived from its gauge, is a major concern. In a karst system, a distance of >10Km is too large to consider a unique flow path with invariable discharge rate all the way to the coastline. Is there any flux measurement in the literature of Pantai Baron with a flow meter to be compared with the flow in Bribin Sindon? Even one measurement could give you an idea of how acceptable this assumption is.

2) A second concern is the presence of other springs along the shoreline that were not considered in this study. In section 5.3 the authors mention that other small submarine and coastal springs are present in the area. Did you identify all of them? Where are they located? I understand that the two main points of discharge are Baron and Ngrumput, but when summed together the smaller springs could represent an important portion of the groundwater discharge and nutrient fluxes in the area. They could also be included in Fig. 1. Why was Pantai Sundak not included in this study if Sir MacDonald and Partners (1984) measured a higher flow here than in Pantai Ngrumput? Were the measurements by Sir MacDonald and Partners (1984) taken during the dry

or wet season? Is it possible that the flux measured in Bribin Sindon feeds not only Pantai Baron but also Pantai Ngrumput and the smaller springs not considered in the study? Including this is in the discussion would improve this issue. The authors mention that a general connection between Pantai Ngrumput and the aquifer system was deduced from the hydrochemistry temporal variability. How was this done exactly? Furthermore, in this type of limestone diffuse discharge is also possible through the matrix. Can this also be occurring in the study area? In Fig. 3 low SST variability can also be observed along the shoreline from Pantai Baron to Pantai Ngrumput. How do you explain this?

Minor comments:

1) The authors mention in section 3.1 that the thermal infrared results were validated by offshore in-situ EC and temperature measurements in November of 2015 and April of 2016, however, no data is presented from these surveys. I suggest to include these data and explain the trends in the results section.

2) The potential impact of excess nutrients in the coastal ocean (such as HABs) is mentioned several times in the manuscript. Has any of this issues been reported in the study area in the past by previous studies? It would be of great interest to mention in the discussion section the specific ecological implications that may arise in this particular area. For instance, is there any vulnerable biota or seagrass species... in the area?

Technical corrections:

Page 2, L 16: I suggest changing "backed" with "supported".

Page 3, L 1: I suggest adding "the" before "dry season" here and throughout the manuscript. The same for the rainy season.

Page 3, L 24: there is a typo, it should be "A decrease" not "An increase".

Page 5, L 5: I suggest mentioning the lab at which the samples were measured and delete "In Germany".

Page 5, L 19-21: this information was already included in page 3, L 23-27.

Page 5, L 29-31: this information was already included in page 4, L 3-5.

Page 6, L 1-4: the description of Pantai Baron can be better explained. I suggest changing the part where you distinguish between the near shore area and the 500 m away from the shore.

Page 7, L 6: I suggest using mol/day as used in the figures instead of "mol per day". Please correct elsewhere.

Page 7, L18-19: this information is repeated, is it necessary to remind the reader?

Page 8, L 10: Please change "time" by "season".

Page 8, L 30: be consistent, is it "Urea" or "urea"?

Figures:

Figure 1: to improve the figure you could superimpose the ESRI World Shaded Relief layer (partially transparent) to give an idea of the topography in the area for an easier understanding. It would also be helpful to include general groundwater flow lines to indicate at least the discussed hydrogeology if possible.

Figure 3: please add a scale bar near the north arrow for reference.

Figure 4: in figures 1, 2, 3, and 5 you used a different font, please be consistent and use the same font here too. I also suggest to change the axes range so the reader can see the data better. You could plot 2H only from -55 to -30 and 18O from -7.5 to -5.0.

Figure 6: please be consistent and use the same font here too. I also suggest to change the bars order to follow the legend, where the nitrate fluxes bar would be placed first followed by the silicate flux and lastly Bribin Sindon discharge.

[Figure]

---

## Referee Comment (RC2) · Anonymous Referee #2 · 16 Jan 2018

This paper focuses on the nutrient fluxes to the coast associated with groundwater discharge in the karstic Gunung Kidul region of Indonesia. The stated aims are twofold; 1) to identify groundwater recharge and flow to the coastal zone, and 2) to elucidate temporal variation in nutrient fluxes to the coast due to groundwater discharge. This subject matter is suitable for publication in HESS, but unfortunately neither of these aims is well addressed in the current manuscript. Robust conclusions are limited by a) assumptions that are not justified, and b)reliance on relatively sparse data on groundwater flow rates (high temporal resolution but only at one point in the groundwater system, 10km from the coast) and nitrate concentrations (at multiple spatial locations but temporally sparse) without adequate consideration of the inherent uncertainty in their

approach. In the introduction, the stated novelty of this study was to capture temporal behavior, but in the closing statement of the conclusion, the authors themselves seem to be acknowledging that the data set presented doesn't adequately capture temporal variability. The authors should reconsider the novel contribution that can be reasonably made using these data before resubmitting their manuscript.

Specific comments

This first aim is very similar to a 2016 paper published by one of the authors (Eiche et al 2016) and it seems that perhaps this aim has already been addressed in the previous paper. If additional novel insights are provided in this new manuscript they should be more clearly highlighted. While precipitation data are presented, recharge is not explicitly estimated, and subsurface flow to the coast was only measured at one location within the karst system. The authors seem to assume that discharge measured within the subsurface is approximately equal to, or at least correlated to recharge, which is probably a reasonable assumption, but this is not clearly stated. What are the implications of "piston flow", as reported in Eiche 2016, on the assumptions made in this paper, does this change the time lag/concentration relationships relative to rapid transport of "new" event water through conduits?

The major reported finding is that nitrate fluxes to the coast are highest during heavy rainfall after a dry period, when both groundwater discharge and nitrate concentrations are high. The temporal resolution of the data, and the data gap in discharge measurements when highest NO3 was measured, makes it difficult to justify strong conclusions. The temporal resolution during the recession period April-July is good, and supports the interpretation, but the rest of the record is arguably too patchy to make strong conclusions about NO3 concentrations during high flow events. The increase in nitrate concentrations during the Dec 2016 rainfall event, and decreases in nitrate during the dry period May-July 2016 event do seem to support the conclusion that nitrate fluxes to the coast are highest during heavy rainfall after a dry period, when both groundwater discharge and nitrate concentrations are high. However, the highest nitrate concen-

trations were actually measured during a period of approx. average discharge in Sept 2016 (there is a gap in the measured discharge time series at immediately prior to the measurement of these peak concentrations). Correlation and trend statistics are not presented, but the authors report both positive and negative correlations between discharge and nutrient concentrations, which seems to suggest that across the data set, there is actually no correlation. A more robust statistical treatment should be presented to support the authors' interpretation of the data and justify their conclusions.

Dissolved silica and nitrate fluxes to the coast associated with groundwater discharge are estimated by multiplying snapshot measurements of Si and NO3 concentrations at the coastal springs with groundwater flow rates measured at one location inland on either 4 or 14 days prior to the measurement of nutrient concentrations. These time lags (<4 and 14 days) are assumed based on tracer studies reported in a report published in the 1980's, and a tracer test conducted in 2012. It is not clear the extent to which groundwater flow conditions during these previous studies relate to the current study. Do the results of Eiche et al. 2016 not provide more recent insights? Regardless of the time lags used, the assumption that discharge measured approx. 10 km inland of the coast on one specific day and at one location is adequate to quantify the groundwater discharge rate at the coast 4 or 14 days later seems an oversimplification. At a minimum some attempt should be made to quantify the uncertainty in the calculated solute fluxes.

It is also not clear exactly what was sampled at the coastal springs. The authors report that these samples were brackish, suggesting these samples were a mix of groundwater discharge and seawater. In which case concentrations measured in these samples would reflect a mixture of these to end memebers. This seems to be what the authors are referring to on Pg 5 when they say brackish values were "normalized" to freshwater according to Hunt and Rosa 2009. Looking at the cited report, it seems likely that the authors did a mixing calculation to work out the concentration in the groundwater component of their brackish samples. If this is the case then the calculation and values used need to be explicitly stated and uncertainty in this calculation quantified and propagated through the analysis.

Some of the data that is presented does not seem to link to the stated aims of the paper. In addition to precipitation, discharge and concentration data, the authors also present data on stable isotopes of water and sea surface temperatures. These data sets do not seem to be well linked to the stated aims and for the stable isotopes in particular, their inclusion in the manuscript could be reconsidered. The sea surface temperature data do show areas of low variability along the coast that the authors interpret as points of groundwater discharge. However, these zones of discharge seem to have been previously identified and named, so this qualitative confirmation of their location seems to be of limited value in addressing the stated aims. The interpretation of stable isotopes data is relatively superficial, and does not meaningfully link to either recharge processes or discharge fluxes. A link between the amount of rainfall associated with recharge events and stable isotopic composition could be expected (i.e. more depleted values in larger rainfall events), but this is not discussed. Increased temporal resolution of stable isotope data may have provided confirmation of time lags between groundwater flow at the subsurface discharge measurement point and groundwater discharge at the coast. However, given the resolution of the stable isotope data presented (∼monthly at best), the value of these data in addressing the stated aims seems minimal. The same could arguably be said for the dissolved silica data (fluxes are calculated by their significance to the aims of the paper is not clear), Concentrations of nutrients other than NO3 are also presented in Table 2, but not discussed in any depth and fluxes are not calculated. Similarly, pH and DO data are reported in Table 2, but don't seem to be used in the analysis.

Technical corrections

Pg1

Abstract: L19 The timescale of recharge and transport to the coast is not explicitly

measured in this study, it is assumed from previous work.

L23 Dsi is not defined. L24 Consider rephrasing to avoid the word "counterintuitive", this seems to demonstrate that dilution is not a dominant control on nitrate fluxes to the coast.

Q. How do these estimated nutrient fluxes to the coast compare to river outflow and runoff, or submarine groundwater discharge?

Pg2

L4-6 References required, and there seems to be more than one idea in here: one is about groundwater travel times, the other is about nutrient retention rates. And do you actually mean in the aquifer, or do you mean in the soil/unsat zone? It's not clear by what mechanism nutrients would be retained in the aquifer under high discharge rates.

L12 "despite of the" needs changing, and L14 "at the example" needs changing

L16 I wouldn't agree that 1 years worth of data constitutes a "long term data set".

L33 The Gunung Sewu area is not clearly defined on Fig 1, the word is labelled, but where is the boundary?

Pg3

L6-7 What is an "underground full dam"? Is it simply a cavity in the limestone that is below the watertable? Is it filled with sand? What makes it a "dam"? And presumably 75000 people, not 75

L15 Nanonyo 2014 seems to be a PhD thesis (thought this is not written explicitly as such in the reference list). It would be better to cite the published journal articles that came out of the research work, rather than the thesis.

L19 For clarity I suggest the authors use something like "subsurface flow" to refer to water flowing in the subsurface, and restrict the use of "discharge" to the actual discharge

of groundwater at the coast. Water flow within an aquifer is not generally referred to as "discharge", in a groundwater context "discharge" usually refers to water leaving the aquifer, as the opposite of "recharge".

L29-32 Where are these branches on the site map? If the major subsurface flow paths have been mapped these should be shown on Fig 1.

L33-7 This seems to imply that groundwater flow within the matrix only happens during low-rainfall periods, which is not the case. Groundwater flow within the matrix would be continuous, but small relative to the amount of groundwater flowing along conduits following rainfall events. Also, be careful to be clear on the two separate processes of recharge and groundwater flow, the two seem to be used here almost as if they are the same thing. The relevance of the water quality comments to end this section to the study is not immediately clear, expansion of this paragraph may be helpful.

Pg4

L4 Multiple lines of evidence have been used in this study, but it doesn't seem like multiple methods were actually "compared". This implies that the same types of estimates (i.e. discharge volume) came out of each method and these values were compared, but this is not actually the case.

L15 This sentence needs rewording.

L17 It seems as though the location of groundwater discharge at the coast was identified prior to this study, the authors should clarify exactly what the new contribution of this SST data is relative to what was known previously.

L24 The relevance of the Siebert 2014 reference here is not clear. "groundwater uninfluenced" is a rather clunky way of putting it.

L28 Figure 3 is referred to prior to Figure 2 (not cited until pg 6)

Pg5
L1-4 What is the relevance of the pH, and DO data to the manuscript?

L5 Delete "In Germany" and correct "isotopy"

L8 Are the nutrients also dissolved? Or ar these total NO3 etc.? If you filtered then aren't these dissolved? And why didn't you estimate fluxes of all nutrients, why only Si and NO3?

Section 3.3 The assumptions made in this section do not seem overly simplistic, as mentioned above. Event-scale variation in stable isotopic values may be able to back up/test these assumptions – i.e. depleted signature during heavy rainfall.

Section 4.1 First paragraph is not results and is mostly repeated from earlier in the manuscript.

Pg6 L3-19 The value of this detailed description of precip data to the manuscript is not clear. A more concise treatment may be to calculate the time lags between precip and groundwater flow or discharge at the coast, rather than a full description of each event, which can be seen on Fig 2 anyway.

L22-23 "to which it is bounded" isn't quite right. Avoid the use of the word "shows", as it is not the correct word.

L28-32 The relevance of this stable isotope analysis to either groundwater recharge, flow or discharge at the coast is not clear.

L34-35 Why define DSi? Why not just use Si – NO3 is also dissolved isn't it? Fig 5A+B – just call it Fig 5 (it only has the two parts).

L36-37 Avoid using terms relative terms like similar (without specifying what it is similar to) and high (what exactly does "high temporal variability" mean?)

Pg7 L5-15 Can you do some stats to back up your interpreted relationships? Are NO3 and Si correlated with subsurface flow? Or are there lag times between peak NO3 and peak flow rates? Also, what is the value of the Si data in this analysis? It doesn't seem

link back to your stated aims.

Discussion: The discussion contains references to a number of figures, which suggests that these comments should have been made in the results section. The discussion should not present or highlight new information about the data that wasn't already presented in the Results section.

L19 Avoid general terms such as "a major amount" and "A further part".

L22 A lag time of 4 days between precip and groundwater flow is not self-evident from Fig 2, and why was this not highlighted in the results section? Previously you mentioned the lag times as having been assumed from the results of previous studies.

L25-27 (and Figure 4) Given rapid infiltration and recharge during rainfall events, why do the stable isotope values not plot on the local meteoric water line?

L30-37 This discussion seems tangential to the current study. You have said in your introduction that rainfall events increase turbidity in the subsurface, and referenced your earlier paper. It doesn't seem like this paper has contributed anything new to our understanding of E. Coli or tourism. The majority of 5.1 seems to have already been covered by Eiche et al 2016.

Pg8 L4-10 This paragraph discusses correlation between data sets, but no correlation statistics were reported in the results section. What does it mean to have both positive and negative correlation? Does this mean there actually isn't a correlation if you look at the full data set?

L11-19 The relevance of the Si data and interpretation to the stated aims of the paper are not clear. You say here that the Si concentrations are diluted during low flow events, does this then support your interpretation that $NO_3$ stores must be released from the unsaturated zone during floods? On Fig 5, during the recession period where you actually have good temporal resolution of data, Si increases while $NO_3$ decreases, what does this mean in terms of process? You write "a further DSi source" do you

mean further spatially, or an additional source? The relevance of the comments on colloidal transport to the current study is not clear.

L20-35 Delete "from these fertilizers" at L30. What is "temporal exhaust" on L 35?

Pg9 L1-3 This seems more like an introductory statement. Correlation is mentioned again, but stats not reported.

Section 5.3 The treatment of uncertainties is inadequate given the assumptions in the analysis (see comments above). This section identifies some sources of error, but does not actually report any quantified uncertainties. L9-10 What do you mean by "A general connection. . ..was deduced from temporal variability of hydrochemistry"?

L18-21 The conclusion begins by acknowledging that a vast area of hinterland may contribute to nutrient discharge at the coast, so it is not clear how does the spatially sparse data set (on groundwater sampling location) can provide a robust estimate of nutrient fluxes. L28 What is "highly variable"?

L32-35 (and L1-2 Pg 10) This seems to be suggesting a better sampling design for the current study, to capture temporal variability by measuring at a higher temporal resolution.

References: Eiche 2016 title is incomplete.

Fig 1. There is a light blue colour in the mid-left of the map that doesn't seem to be explained in the legend.

Fig 2 Why are the discharge data so patchy?

Fig 4 X-type symbols are too similar.

Fig 6 Is the discharge shown on the day of the flux estimate, or the day used to calculate the flux estimate? Ie the discharge on the day NO3 concs were measured, or the discharge 4-14 days prior? Why use a bar chart instead of a time series?

[Figure]

Table 1 Use delta, not d for isotopes. NO2, NH4 and PO4 data are not discussed in any detail in the manuscript. Nutrients and NO3 are not the same thing. Is it a paper on all of these nutrients, or just on nitrate fluxes?

Table 2 Caption says average discharge rates, but table reports avg (presumably) as well as min and max, average and standard deviation would be more concise. Table should indicate which are dry season and wet season samples (=4 or 14 days prior to concentration measurement). Some uncertainty on the flux estimates should be provided. Why are fluxes of NH4 and PO4 not provided? Measured concentrations used to calculate fluxes should also be reported in this table.

---

## Author Comment (AC1) · 16 Feb 2018

Dear Editor,

we thank reviewer 1 for the fruitful comments on the manuscript. The major remark of reviewer 1 was that there are probably more groundwater flow paths within the karstic system. We addressed this issue and collected all information about known flow paths, as well as all negotiated flow paths, in the karstic system (Figure 1, Table 1). Based on these flow paths we give a range of nutrient fluxes at the main outlet into the ocean (Pantai Baron) in Figure 5 and Table 4. Below you can find a detailed point to point reply to the comments made by reviewer 1.

[Figure]

Best regards,

Till Oehler

Reviewer

I agree with the authors that the main uncertainties in this study are the connection between the catchment area and the nutrients discharge to the coastal ocean. These two points are basically the main goals of the study and a better job must be done to justify the lack of data in this concern: As the authors mention, assuming that the discharge in Pantai Baron is the same as the flow measured in Bribin Sindon and can be directly derived from its gauge, is a major concern. In a karst system, a distance of >10Km is too large to consider a unique flow path with invariable discharge rate all the way to the coastline. Is there any flux measurement in the literature of Pantai Baron with a flow meter to be compared with the flow in Bribin Sindon? Even one measurement could give you an idea of how acceptable this assumption is.

Answer

In the previous version of the manuscript we discuss under section "5.3 uncertainties" that additional flow paths may occur, and we agree with the reviewer's suggestion that this issue can be resolved in a better way. We now include all connections which were proven by tracer tests, as well as all suspected connections, and discharge rates into Figure 1 and Table 1 and show these results in a revised version in section "2.2 Subsurface Hydrology". Based on these datasets we give a range of groundwater nutrient fluxes (Figure 5, Table 4). In general, the discharge range which was reported at Pantai Baron from McDonald&Partners 1984 is in a similar range than measured at Bribin and our data suggest that Bribin-Baron is major pathway of groundwater flow. We can assume that all water which passes by the subsurface river dam flows towards the ocean and based on all known pathways, we can also assume that most of this water discharges at Baron.
Reviewer

A second concern is the presence of other springs along the shoreline that were not considered in this study. In section 5.3 the authors mention that other small submarine and coastal springs are present in the area. Did you identify all of them? Where are they located? I understand that the two main points of discharge are Baron and Ngrumput, but when summed together the smaller springs could represent an important portion of the groundwater discharge and nutrient fluxes in the area. They could also be included in Fig. 1. Why was Pantai Sundak not included in this study if Sir MacDonald and Partners (1984) measured a higher flow here than in Pantai Ngrumput?

Answer

All springs which we were able to identify in the area are springs Baron, Ngrumput, Slili, Sundak, Ngobaran, and Pok Tunggal (Figure 1, Table 1). All known discharge rates from these springs summed up show still a much lower discharge than at Pantai Baron. Furthermore a qualitative tracer test between Seropan and Baron indicated only a connection between these two systems and not towards the springs Slili, Sundak, Ngobaran and Pok Tunggal. A connection between Pantai Ngrumput has not been considered yet, but our nitrate data at Pantai Baron and Patani Ngrumput correlates (Spearman's rank) with each other (Table 3) indicating that both springs are fed by a similar groundwater water the hinterland.

Reviewer

Were the measurements by Sir MacDonald and Partners (1984) taken during the dry or wet season?

Answer

The measurements by Sir MacDonald and Partners (1984) were taken during the dry season in August

Reviewer

[Figure]

Is it possible that the flux measured in Bribin Sindon feeds not only Pantai Baron but also Pantai Ngrumput and the smaller springs not considered in the study? Including this is in the discussion would improve this issue.

Answer

It is possible that Bribin-Sindon feeds also further springs at the coast, but we cannot say this for sure. We show all known pathways (and known non-pathways) which have been mapped in the area in Figure 1 and Table 1. However, the essential aim of this manuscript is to show the temporal variability of nutrient fluxes, not so much the spatial variability.

Reviewer

The authors mention that a general connection between Pantai Ngrumput and the aquifer system was deduced from the hydrochemistry temporal variability. How was this done exactly?

Answer

A Spearman's rank correlation (Table 3) of nitrate concentrations between the different springs indicates that nitrate shows a similar variability in concentrations in between Pantai Baron, Pantai Ngrumput and Gunung Kendil. This indicates that similar processes force variations in nitrate concentrations in between different springs. In a revised version we resolve this issue in more detail.

Reviewer

Furthermore, in this type of limestone diffuse discharge is also possible through the matrix. Can this also be occurring in the study area?

Answer

Matrix flow is as in any karstic region present all the time. Matrix flow is responsible for a baseline signal of the physio-chemistry of the groundwater as shown in Eiche et

al 2016). However, this component is relatively small and slow compared to the piston flow. Especially for nutrients the extrema are for sure controlled by the piston flow. We can include this in the discussion of a revised paper.

Reviewer

In Fig. 3 low SST variability can also be observed along the shoreline from Pantai Baron to Pantai Ngrumput. How do you explain this?

Answer

This might be caused by longshore coastal currents and indicates the extent to which groundwater discharge affects the coastal ocean. We include this information in section "4.1 Groundwater recharge, flow, and discharge to the coastal ocean"

Minor comments

Reviewer

1) The authors mention in section 3.1 that the thermal infrared results were validated by offshore in-situ EC and temperature measurements in November of 2015 and April of 2016, however, no data is presented from these surveys. I suggest to include these data and explain the trends in the results section.

Answer

We include this data into a revised version of the manuscript and explain the trends in the results section (Figure 2).

Reviewer

2) The potential impact of excess nutrients in the coastal ocean (such as HABs) is mentioned several times in the manuscript. Has any of this issues been reported in the study area in the past by previous studies? It would be of great interest to mention in the discussion section the specific ecological implications that may arise in this particular

area. For instance, is there any vulnerable biota or seagrass species in the area?

Answer

Seagrass species were observed in a fringing reef at Pantai Ngrumput. Furthermore fishing is an important economy in the area. We can include information on the seagrass and the reef in the area, and some information about fishing.

Technical corrections

Page 2, L 16: I suggest changing "backed" with "supported".

-Agreed for a revised version.

Page 3, L 1: I suggest adding "the" before "dry season" here and throughout the manuscript. The same for the rainy season.

-Agreed for a revised version.

Page 3, L 24: there is a typo, it should be "A decrease" not "An increase".

-Agreed for a revised version.

Page 5, L 5: I suggest mentioning the lab at which the samples were measured and delete "In Germany".

-Agreed.

Page 5, L 19-21: this information was already included in page 3, L 23-27.

- We will reformulate this in a revised version.

Page 5, L 29-31: this information was already included in page 4, L 3-5.

- We will reformulate this in a revised version.

Page 6, L 1-4: the description of Pantai Baron can be better explained. I suggest changing the part where you distinguish between the near shore area and the 500 m

away from the shore.

-Agreed. We will change this in a revised version

Page 7, L 6: I suggest using mol/day as used in the figures instead of "mol per day". Please correct elsewhere.

-Agreed. This will be changed in a revised paper.

Page 7, L18-19: this information is repeated, is it necessary to remind the reader?

-We can leave this information out in a revised manuscript.

Page 8, L 10: Please change "time" by "season".

-We will change this part as suggested by reviewer 2 as well.

Page 8, L 30: be consistent, is it "Urea" or "urea"?

-We will stick to urea and change it accordingly.

Figures:

Figure 1: to improve the figure you could superimpose the ESRI World Shaded Relief layer (partially transparent) to give an idea of the topography in the area for an easier understanding. It would also be helpful to include general groundwater flow lines to indicate at least the discussed hydrogeology if possible.

-We included the ESRI World Shaded Relief layer and vectors with discharge rates and direction into the map in order to clarify the known pathways and volumes of ground-water in the area as suggested by reviewer 1 and reviewer 2 (see Figure 1 and Table 1).

Figure 3: please add a scale bar near the north arrow for reference.

-Agreed

Figure 4: in figures 1, 2, 3, and 5 you used a different font, please be consistent and

use the same font here too. I also suggest to change the axes range so the reader can see the data better. You could plot 2H only from -55 to -30 and 18O from -7.5 to -5.0.

-We can change this accordingly

Figure 6: please be consistent and use the same font here too. I also suggest to change the bars order to follow the legend, where the nitrate fluxes bar would be placed first followed by the silicate flux and lastly Bribin Sindon discharge.

-We show a new bar graph with minimum and maximum fluxes (Figure 5).

**Legend**

| | |
|---|---|
| | Limestone |
| | Alluvium |
| | Volcanic breccia, lava, tuff |
| | Marl, Tuff |
| | Tuff, Sandstone |
| | Sandstone Claystone |
| | Marl, Limestone |
| ◇ | Climate station |
| ● | Coastal spring |
| ○ | Subsurface river |
| ■ | Subsurface river dam |
| ┄┄┄ | NoFlowPaths |
| ─·─► | FlowPaths |

**Fig. 1.** All proven land-ocean groundwater connections (black lines) and negotiated connections (red lines) in the karstic region of Gunung Kidul. For respective discharge rates and names and types o

**Fig. 2.** TIR image of the coastal ocean showing two major sites of groundwater discharge and related electrical conductivity values which were measured in the coastal water.

[Figure]

**Fig. 3.** Discharge at the subsurface river dam (grey) and precipitation data (blue) from the upstream located climate station Ponjong, nitrate concentrations at the coastal spring Pantai Baron (red dots). The

**Fig. 4.** Scatterplot of DSi, NO3, and PO4 in relation to the minimum discharge at Pantai Baron.

[Figure]

Fig. 5. Range of land-ocean groundwater nutrient fluxes estimated based on groundwater discharge rates from a subsurface river dam and nutrient concentrations sampled at Pantai Baron. The upper white part of

| Site | Map ID | Type | Discharge (m³/s) dry season | Discharge (m³/s) wet season | Comment | Reference |
|------|--------|------|-----------------------------|-----------------------------|---------|-----------|
| P. Baron | a | Coastal spring | 4-8.2 | | | 1 |
| P. Ngrumput | b | Coastal spring | 0.05-0.06 | 0.03 | | 2 |
| P. Slili | c | Coastal spring | 0.05 | | No connection to Bribin-Sindon | 1 |
| P. Sundak | d | Coastal spring | 0.2 | | No connection to Bribin-Sindon | 1 |
| Pok Tunggal | e | Coastal spring | | | No connection to Bribin-Sindon | |
| P. Ngobaran | f | Coastal spring | | | No connection to Bribin-Sindon | |
| Buhputih | g | Subsurface river | 0.02 | | Flows to Baron | |
| Bribin-Sindon | h | dam | >1 | <12 | Flows to Baron | |
| Gua Bribin | h | Subsurface river | 1-1.3 | 4-8 | Gua Bribin is 2 km upstream of Bribin-Sindon | 1,3 |
| Seropan | i | Subsurface river | 0.4-0.5 | 0.5 to <3, extreme >10 | Flows via Ngreneng to Baron | 3,4 |
| Grubug | j | Subsurface river | 0.7-1 | 2 | 100% flows to Baron 25% of discharge of Baron | 1,3 |
| Gua Ngreneng | k | Subsurface river | <0.1 | 0.2 | | 1,3 |
| Luweng Jomblangan | l | Subsurface river | | | Flows to Bribin-Sindon | 1,3 |
| Gilap | m | Subsurface river | 0.003 | | Flows to Bribin-Sindon | 1,3 |

* 1 = MacDonalds&Partners 1984; 2 = own measurements 2016; 3 = own measurements 2000/2001; 4 = own measurements 2008-2010

**Fig. 6.** Table 1: All known discharge rates measured at subsurface rivers in the hinterland and coastal springs are shown in this table. The site where the measurement was taken (Flow ID) is shown in Figure 1.

| Event | Date | Season | DO (%) | EC (µS/cm) | Temp (°C) | NO$_3$ (µmol/L) | NO$_2$ (µmol/L) | NH$_4$ (µmol/L) | DSi (µmol/L) | PO$_4$ (µmol/L) |
|---|---|---|---|---|---|---|---|---|---|---|
| Pantai Baron-1 | 14-Nov-2015 | Dry | 83 | 557 | 27.9 | 170 | 0.0 | | 408 | 0.1 |
| Pantai Baron-2 | 19-Apr-2016 | Wet | 85 | 429 | 27.6 | 114 | 0.3 | 1.3 | 335 | 0.1 |
| Pantai Baron-3 | 10-May-2016 | Dry | 88 | 521 | 28.0 | 90 | 0.2 | 0.6 | 458 | 0.1 |
| Pantai Baron-4 | 24-May-2016 | Dry | 82 | 541 | 28.1 | 21 | 0.4 | 3.6 | 471 | 0.1 |
| Pantai Baron-5 | 8-Jun-2016 | Dry | 82 | 525 | 28.0 | 78 | 0.3 | 2.4 | 436 | 0.1 |
| Pantai Baron-6 | 21-Jun-2016 | Dry | 77 | 384 | 27.2 | 27 | 0.2 | 1.2 | 297 | 0.2 |
| Pantai Baron-7 | 20-Aug-2016 | Dry | | 640 | 27.8 | 271 | | | | |
| Pantai Baron-8 | 23-Sep-2016 | Dry | | 820 | 28.4 | 115 | | | | |
| Pantai Baron-9 | 30-Nov-2016 | Wet | | 260 | 23.0 | 52 | | | | |
| Pantai Baron-10 | 13-Dec-2016 | Wet | 92 | 429 | 27.6 | 140 | 0.4 | | 312 | 0.9 |
| Gua Pindul-1 | 21-Apr-2016 | Wet | 92 | 533 | 28.6 | 94 | 0.6 | 1.9 | 320 | 0.1 |
| Gua Pindul-2 | 11-May-2016 | Dry | 88 | 540 | 28.5 | 48 | 0.3 | 1.9 | 389 | 0.1 |
| Gua Pindul-3 | 24-May-2016 | Dry | 87 | 567 | 27.9 | 72 | 0.3 | 1.4 | 413 | 0.1 |
| Gua Pindul-4 | 8-Jun-2016 | Dry | 86 | 558 | 28.4 | 33 | 0.3 | 2.1 | 410 | 0.1 |
| Gua Pindul-5 | 21-Jun-2016 | Dry | 86 | 413 | 27.3 | 34 | 0.4 | 3.3 | 303 | 0.1 |
| Gua Pindul-6 | 20-Aug-2016 | Dry | | 520 | 29.6 | 302 | | | | |
| Gua Pindul-7 | 23-Sep-2016 | Dry | | 560 | 29.0 | 0 | | | | |
| Gua Pindul-8 | 30-Nov-2016 | Wet | | 270 | 25.6 | 0 | | | | |
| Gua Pindul-9 | 12-Dec-2016 | Wet | 99 | 494 | 26.1 | 84 | 1.1 | | 316 | 0.1 |
| Gunung Kendil-1 | 17-Apr-2016 | Wet | 93 | 567 | 27.8 | 123 | 0.5 | 4.1 | 495 | 0.1 |
| Gunung Kendil-2 | 11-May-2016 | Dry | 84 | 522 | 27.4 | 91 | 0.0 | 0.0 | 462 | 0.1 |
| Gunung Kendil-3 | 24-May-2016 | Dry | 82 | 524 | 27.2 | 53 | 0.2 | 1.4 | 497 | 0.1 |
| Gunung Kendil-4 | 8-Jun-2016 | Dry | 73 | 527 | 27.0 | 66 | 0.2 | 0.2 | 474 | 0.1 |
| Gunung Kendil-5 | 21-Jun-2016 | Dry | 82 | 532 | 27.1 | 64 | 0.0 | 0.0 | 450 | 0.1 |
| Kali Suci-1 | 21-Apr-2016 | Wet | 104 | 426 | 29.1 | 58 | 0.2 | 1.1 | 308 | 0.1 |
| Kali Suci-2 | 10-May-2016 | Dry | 102 | 431 | 28.1 | 107 | 0.3 | 1.7 | 362 | 0.1 |
| Kali Suci-3 | 24-May-2016 | Dry | 101 | 485 | 27.5 | 66 | 0.2 | 0.9 | 408 | 0.1 |
| Kali Suci-4 | 8-Jun-2016 | Dry | 104 | 490 | 27.5 | 133 | 0.2 | 0.6 | 382 | 0.1 |
| Kali Suci-5 | 21-Jun-2016 | Dry | 102 | 417 | 26.7 | 69 | 0.1 | 0.3 | 307 | 0.1 |
| Kali Suci-6 | 20-Aug-2016 | Dry | | 520 | 28.9 | 230 | | | | |
| Kali Suci-7 | 23-Sep-2016 | Dry | | 490 | 29.2 | 0 | | | | |
| Kali Suci-8 | 30-Nov-2016 | Wet | | 200 | 26.7 | 38 | | | | |
| Pantai Ngrumput-1 | 16-Nov-2015 | Dry | 75 | 8380 | 27.8 | 132 | 0.0 | | 372 | 0.5 |
| Pantai Ngrumput-2 | 19-Apr-2016 | Wet | 72 | 6300 | 28.4 | 30 | 0.1 | 7.9 | 350 | 0.1 |
| Pantai Ngrumput-3 | 10-May-2016 | Dry | 83 | 9530 | 28.8 | 17 | 0.2 | 7.4 | 368 | 0.1 |
| Pantai Ngrumput-4 | 21-Jun-2016 | Dry | 72 | 9450 | 28.4 | 7 | 0.4 | 5.1 | 391 | 0.1 |
| Pantai Ngrumput-5 | 20-Aug-2016 | Dry | | 7520 | 28.2 | | | | | |
| Pantai Ngrumput-6 | 23-Sep-2016 | Dry | | 7510 | 28.5 | | | | | |
| Pantai Ngrumput-7 | 30-Nov-2016 | Wet | | 8080 | 27.2 | | | | | |
| Pantai Ngrumput-8 | 13-Dec-2016 | Wet | 67 | 5950 | 28.2 | 145 | 0.0 | | 302 | 1.0 |

☐ = base flow at Pantai Baron
▨ = high discharge event at Pantai Baron

**Fig. 7.** Table 2: The hydrochemistry of the springs which are located in the hinterland and at the coast. High discharge events at Pantai Baron are marked by the grey shaded areas.

| | P. Ngrumput | Gunung Kendil | Kali Suci | Goa Pindul |
|---|---|---|---|---|
| P. Baron | 0.90 | 1.00 | 0.12 | 0.44 |
| P. Ngrumput | | 1.00 | -0.50 | 0.80 |
| Gunung Kendil | | | -0.10 | 0.30 |
| Kali Suci | | | | 0.54 |

**Fig. 8.** Table 3: Correlation matrix (Spearman's rank) of temporal NO3 concentration variations of the different springs which were sampled in Gunung Kidul.

| Date | P. Baron min discharge (m³/sec) | P. Baron max discharge (m³/sec) | NO3 (mol/m³) | NO3 min flux (10^6 mol/day) | NO3 max flux (10^6 mol/day) | DSi (mol/m³) | DSi min flux (10^7 mol/day) | DSi max flux (10^7 mol/day) | PO4 (mol/m³) | PO4 min flux (10^4 mol/day) | PO4 max flux (10^4 mol/day) |
|---|---|---|---|---|---|---|---|---|---|---|---|
| Nov-15 | 4 | 5 | 170 | 63 | 74 | 408 | 15 | 18 | 0 | 5 | 6 |
| Apr-16 | 10 | 13 | 114 | 94 | 129 | 335 | 28 | 38 | 0 | 12 | 17 |
| May-16 | 1 | 8 | 90 | 11 | 65 | 458 | 6 | 33 | 0 | 2 | 10 |
| May-16 | 4 | 6 | 21 | 7 | 11 | 471 | 15 | 23 | 0 | 5 | 7 |
| Jun-16 | 2 | 4 | 78 | 16 | 30 | 436 | 9 | 17 | 0 | 2 | 4 |
| Jun-16 | 3 | 8 | 27 | 6 | 18 | 297 | 7 | 20 | 0 | 4 | 13 |
| Aug-16 | 5 | 5 | 271 | 111 | 123 | | | | | | |
| Sep-16 | 5 | 5 | 115 | 46 | 53 | | | | | | |
| Nov-16 | 5 | 13 | 52 | 22 | 59 | | | | | | |
| Dec-16 | 11 | 20 | 140 | 133 | 245 | 312 | 30 | 55 | 1 | 85 | 157 |

☐ = base flow at Pantai Baron

▨ = high discharge event at Pantai Baron

**Fig. 9.** Table 4: Range of groundwater discharge rates, NO3 fluxes, DSi fluxes and PO4 fluxes at Pantai Baron. Flooding events are marked by the grey line.

---

## Author Comment (AC2) · 16 Feb 2018

Dear editor,

we also thank reviewer 2 for the extensive review and think that the manuscript has improved a lot from these suggestions. We would like to stress that we herein present a very unique dataset from a tropical karstic region. To our knowledge, in such regions groundwater nutrient fluxes into the ocean have so far only been studied on a seasonal scale at its best. We present groundwater discharge data with a high temporal resolution of ten minutes over a period of about a year and discuss these data in relation to groundwater nutrient concentrations, which show a large concentration range

(e.g. NO3 from 0 to 300 $\mu$mol/L) (Figure 3). As a consequence groundwater nutrient fluxes show a high temporal variability which has important implications for coastal ecosystems, as well as for the coastal groundwater quality. We follow the reviewer's suggestion and remove the stable isotopes from the manuscript as they do not provide any relevant information. Instead we present a more detailed statistical analysis of our dataset in Figure 4 and Table 3. We also give a range of minimum and maximum nutrient fluxes as suggested by the reviewer (Figure 5, Table 4). We will also remove section "5.3 Uncertainties" in the discussion as we now quantified this error. We will further change the scope of section "5.1 Groundwater recharge and flow towards the coastal ocean". Instead of discussing groundwater recharge, we will discuss the possible nutrient sources in the hinterland and how these nutrients are transported into the aquifer and towards the ocean. We are confident that the manuscript has improved a lot from these reviews and hope that a revised version will be considered for publication in HESS.

Best regards,

Till Oehler

Reviewer

This paper focuses on the nutrient fluxes to the coast associated with groundwater discharge in the karstic Gunung Kidul region of Indonesia. The stated aims are twofold;1) to identify groundwater recharge and flow to the coastal zone, and 2) to elucidate temporal variation in nutrient fluxes to the coast due to groundwater discharge. This subject matter is suitable for publication in HESS, but unfortunately neither of these aims is well addressed in the current manuscript. Robust conclusions are limited by a) assumptions that are not justified, and b)reliance on relatively sparse data on groundwater flow rates (high temporal resolution but only at one point in the groundwater system, 10km from the coast) and nitrate concentrations (at multiple spatial locations but temporally sparse) without adequate consideration of the inherent uncertainty in their approach.

In the introduction, the stated novelty of this study was to capture temporal behavior, but in the closing statement of the conclusion, the authors themselves seem to be acknowledging that the data set presented doesn't adequately capture temporal variability. The authors should reconsider the novel contribution that can be reasonably made using these data before resubmitting their manuscript.

Answer

A groundwater discharge dataset of several years with a temporal resolution of 10 minute intervals is very unique in a remote tropical karstic region such as Gunung Kidul. It might also be important to mention that a recent flood event lead to a severe damage of the dam and such a dataset might not be available any more in the near future. Hydrochemical datasets with a monthly temporal resolution over a period of a year are also extensive, considering that most studies dealing with land-ocean groundwater nutrient fluxes have been carried out on a seasonal scale at best. We therefore think that we have the opportunity to show an extensive dataset in this manuscript and substantially enhanced the data analysis. In a revised version of the manuscript we show all known flow paths and discharge rates in detail. We leave out the isotopic data as suggested by reviewer 2. We also analyze the precipitation and discharge data in a more detailed way, and present statistics about nutrient concentrations and fluxes.

Reviewer

This first aim is very similar to a 2016 paper published by one of the authors (Eiche et al 2016) and it seems that perhaps this aim has already been addressed in the previous paper. If additional novel insights are provided in this new manuscript they should be more clearly highlighted. While precipitation data are presented, recharge is not explicitly estimated, and subsurface flow to the coast was only measured at one location within the karst system. The authors seem to assume that discharge measured within the subsurface is approximately equal to, or at least correlated to recharge, which is probably a reasonable assumption, but this is not clearly stated. What are the implications of "piston flow", as reported in Eiche 2016, on the assumptions made in this paper, does this change the time lag/concentration relationships relative to rapid transport of "new" event water through conduits?

Answer

The focus of Eiche et al 2016 was to get insight into the type of groundwater flow and its dynamics, and give some information about the hydrochemistry in the area in dependence of water type and season. This paper is a significant step forward. It combines the information from Eiche et al 2016 with new data that focus on nutrient transport towards the sea. First of all we identify 4 different sampling events with high precipitation rates and high discharge in April 2016, June 2016, November 2016 and December 2016 (Figure 3, red arrows). These events are also indicated in the physico-chemistry, e.g. indicated by a lower EC and a lower temperature as well as lower DSi concentrations at Pantai Baron (Table 2). We then discuss in a revised version of the manuscript the different processes which may explain these results. For example a heavy rain event at the end of the flood recession period didn't lead to high NO3 concentrations which might be the result of dilution in the aquifer by rainwater, while there was no new NO3 being washed into the aquifer. In Dec-16 high NO3 concentrations might be explained by a high discharge event which also led higher PO4 concentrations and fluxes. DSi is under such a setting also of interest, because karstic regions are in general characterized by low DSi concentrations, while on tropical volcanic island erosion of volcanic lithologies in the hinterland (see Figure 1) may lead to high DSi groundwater concentrations. Our datasets also indicate that DSi is controlled by dilution in the aquifer due to negative linear correlation with discharge, while NO3 is controlled by both a source in the Epikarst and dilution in the aquifer (e.g. see Figure 4).

Reviewer

The major reported finding is that nitrate fluxes to the coast are highest during heavy

rainfall after a dry period, when both groundwater discharge and nitrate concentrations are high. The temporal resolution of the data, and the data gap in discharge measurements when highest NO3 was measured, makes it difficult to justify strong conclusions. The temporal resolution during the recession period April-July is good, and supports the interpretation, but the rest of the record is arguably too patchy to make strong conclusions about NO3 concentrations during high flow events. The increase in nitrate concentrations during the Dec 2016 rainfall event, and decreases in nitrate during the dry period May-July 2016 event do seem to support the conclusion that nitrate fluxes to the coast are highest during heavy rainfall after a dry period, when both groundwater discharge and nitrate concentrations are high. However, the highest nitrate concentrations were actually measured during a period of approx. average discharge in Sept 2016 (there is a gap in the measured discharge time series at immediately prior to the measurement of these peak concentrations). Correlation and trend statistics are not presented, but the authors report both positive and negative correlations between discharge and nutrient concentrations, which seems to suggest that across the data set, there is actually no correlation. A more robust statistical treatment should be presented to support the authors' interpretation of the data and justify their conclusions.

Answer

In a revised version of the manuscript we will give a much more statistical analysis of our dataset. One of the main findings is that we have high nutrient fluxes during wet season in 2015/2016 as well as during the wet season in December 2016 which was followed by a very heavy rain and discharge event. A further strong point, as stated by the reviewer as well, is the flood recession period from April until June 2016 which leads to very low nutrient fluxes, caused by low discharge coupled to very low nutrient concentrations. Low nutrient concentrations were also observed in June 2016 after the flood recession period, indicating that during this time NO3 concentrations in the aquifer were controlled by dilution without a further source in the Epikarst. High nutrient concentrations in August 2016 in turn followed the longest and driest period in

this study. We assume that the extended dry period may be one reason for the high concentrations in August 2016, e.g. accumulation over dry periods and then combined flush in of the nutrients when first rain events occurred. We further provide a more detailed statistical analysis of our dataset. We correlate DSi, NO3 and PO4 with the minimum discharge at Pantai Baron (Figure 4). We can see that a negative linear correlation between discharge and DSi which indicates that a DSi is controlled by dilution from new event water, e.g. such as in April 2016, 21st of June 2016, November 2016 and December 2016 (see Table 1). In comparison, discharge and NO3 did not show such a correlation, indicating that dilution from new event water is only one part which controls the concentrations, but that also NO3 sources from the Epikarst lead to an additional input of NO3. A correlation matrix of NO3 concentrations in between the different springs suggests further that Pantai Baron, Pantai Ngrumput, and Gunung Kendil follow a similar pattern, e.g. NO3 concentrations are controlled by similar processes.

Reviewer

Dissolved silica and nitrate fluxes to the coast associated with groundwater discharge are estimated by multiplying snapshot measurements of Si and NO3 concentrations at the coastal springs with groundwater flow rates measured at one location inland on either 4 or 14 days prior to the measurement of nutrient concentrations. These time lags (<4 and 14 days) are assumed based on tracer studies reported in a report published in the 1980's, and a tracer test conducted in 2012. It is not clear the extent to which groundwater flow conditions during these previous studies relate to the current study. Do the results of Eiche et al. 2016 not provide more recent insights? Regardless of the time lags used, the assumption that discharge measured approx. 10 km inland of the coast on one specific day and at one location is adequate to quantify the groundwater discharge rate at the coast 4 or 14 days later seems an oversimplification. At a minimum some attempt should be made to quantify the uncertainty in the calculated solute fluxes.

Answer

Reviewer 1 had a similar comment, and we now deal with these uncertainties in a revised version of the manuscript (see above). We therefore give a range of groundwater discharge, which is based on different flow paths in the area, but which also considers a time span before sampling. We assume a groundwater travel time which is between 12 and 16 days during non-flooding events and between 2 and 6 days during flooding events. From this time span we calculated a minimum and maximum discharge rate at the subsurface river dam Bribin Sindon. We furthermore add discharge rates from other contributors to Pantai Baron (e.g. Grubug, Kali Suci) as shown in Table 1 and come up with a minimum and maximum groundwater discharge flux. Discharge multiplied with nutrient concentrations from Pantai Baron yield a range of groundwater nutrient fluxes in the region (Figure 5). The results from Eiche et 2016 are based on discharge measurements from the Seropan cave and are related to flow dynamics (piston flow, matrix flow). Groundwater travel times towards the ocean are not presented in this work.

Reviewer

It is also not clear exactly what was sampled at the coastal springs. The authors report that these samples were brackish, suggesting these samples were a mix of groundwater discharge and seawater. In which case concentrations measured in these samples would reflect a mixture of these to endmembers. This seems to be what the authors are referring to on Pg 5 when they say brackish values were "normalized" to freshwater according to Hunt and Rosa 2009. Looking at the cited report, it seems likely that the authors did a mixing calculation to work out the concentration in the groundwater component of their brackish samples. If this is the case then the calculation and values used need to be explicitly stated and uncertainty in this calculation quantified and propagated through the analysis.

Answer

At Pantai Baron freshwater was sampled, so these values were not unmixed. At Pantai

Ngrumput brackish water was sampled, so these values were unmixed in the previous version of the manuscript. However, this unmixing is actually not necessary in the current version of the manuscript and we would therefore remove this calculation in a revised version.

Reviewer

Some of the data that is presented does not seem to link to the stated aims of the paper. In addition to precipitation, discharge and concentration data, the authors also present data on stable isotopes of water and sea surface temperatures. These data sets do not seem to be well linked to the stated aims and for the stable isotopes in particular, their inclusion in the manuscript could be reconsidered. The sea surface temperature data do show areas of low variability along the coast that the authors interpret as points of groundwater discharge. However, these zones of discharge seem to have been previously identified and named, so this qualitative confirmation of their location seems to be of limited value in addressing the stated aims. The interpretation of stable isotopes data is relatively superficial, and does not meaningfully link to either recharge processes or discharge fluxes. A link between the amount of rainfall associated with recharge events and stable isotopic composition could be expected (i.e. more depleted values in larger rainfall events), but this is not discussed. Increased temporal resolution of stable isotope data may have provided confirmation of time lags between groundwater flow at the subsurface discharge measurement point and groundwater discharge at the coast. However, given the resolution of the stable isotope data presented (_monthly at best), the value of these data in addressing the stated aims seems minimal. The same could arguably be said for the dissolved silica data (fluxes are calculated by their significance to the aims of the paper is not clear), Concentrations of nutrients other than NO3 are also presented in Table 2, but not discussed in any depth and fluxes are not calculated. Similarly, pH and DO data are reported in Table 2, but don't seem to be used in the analysis.

Answer

The value of these datasets for the interpretation of our results could be improved in a revised version of the manuscript. The TIR pictures indicate to which spatial extent coastal waters are influenced by groundwater discharge. Some of the springs were known before, but the TIR pictures clearly show how far they may spread outwards into the ocean. We agree with the reviewer that the conclusion we can get from the stable isotopes are very limited, and we would remove these data from a revised version of the manuscript. Dissolved silicate (DSi) is an important nutrient for diatoms in the coastal ocean. The objectives of showing DSi fluxes as well can be stated more clearly in a revised version. In a revised version we would show NO3, DSi and PO4 fluxes. Nitrogen obviously occurred dominantly in the form of NO3, which is in turn forced by an oxic aquifer. We therefore show oxygen values in table 2 and only refer to No3 fluxes in Figure 5.

Technical corrections:

Pg1

Abstract: L19 The timescale of recharge and transport to the coast is not explicitly measured in this study, it is assumed from previous work.

-We can remove this information from the abstract in a revised version of the manuscript.

L23 Dsi is not defined.

-In general we refer to the term DSi as dissolved silicon. We can define this the first time we mention it in the manuscript.

L24 Consider rephrasing to avoid the word "counterintuitive", this seems to demonstrate that dilution is not a dominant control on nitrate fluxes to the coast.

-We can rephrase this sentence.

Q. How do these estimated nutrient fluxes to the coast compare to river outflow and
runoff, or submarine groundwater discharge?

-Characteristic for a karst region is that surface discharge like rivers is rare. There is no known river which discharges into the coastal ocean within the region. Rivers in the hinterland consist of the Oyo river, Betung river and Kali Suci, with the latter one evolving into a sinking stream which becomes a subsurface river and discharges at Baron.

Pg2

L4-6 References required, and there seems to be more than one idea in here: one is about groundwater travel times, the other is about nutrient retention rates. And do you actually mean in the aquifer, or do you mean in the soil/unsat zone? It's not clear by what mechanism nutrients would be retained in the aquifer under high discharge rates.

-We can be more specific and mention in a first sentence that groundwater travel times are quick. As stated, we mean that nutrient retention in the aquifer is short. We can include a reference (e.g. Moosdorf et al 2015).

L12 "despite of the" needs changing, and L14 "at the example" needs changing

-We can change it in a revised version

L16 I wouldn't agree that 1 years worth of data constitutes a "long term data set".

-We can change the wording

L33 The Gunung Sewu area is not clearly defined on Fig 1, the word is labelled, but where is the boundary?

-We included a shaded relief map into Figure 1, in which the mountainous region shows the Gunung Sewu area

Pg3

L6-7 What is an "underground full dam"? Is it simply a cavity in the limestone that is

[Figure]

HESSD

Interactive
comment

below the watertable? Is it filled with sand? What makes it a "dam"? And presumably 75000 people, not 75

-We can add more information here: "At Bribin Sindon, the karst river is dammed up by a concrete barrage, which completely closed the elliptic cross section of the cave, creating an underground water storage which is managed by means of a hydropower-driven pumping system. This hydrowerplant supplies water for more than 75,000 people in the area."

L15 Nanonyo 2014 seems to be a PhD thesis (thought this is not written explicitly as such in the reference list). It would be better to cite the published journal articles that came out of the research work, rather than the thesis.

-A PhD work is an openly published document, in this case also in English. We will explicitly mention this in the reference list.

L19 For clarity I suggest the authors use something like "subsurface flow" to refer to water flowing in the subsurface, and restrict the use of "discharge" to the actual discharge of groundwater at the coast. Water flow within an aquifer is not generally referred to as "discharge", in a groundwater context "discharge" usually refers to water leaving the aquifer, as the opposite of "recharge".

- The definition of discharge is: "the volumetric flow rate of water that is transported through a given cross-sectional." Discharge is measured at the dam, so in our view this is the right term.

L29-32 Where are these branches on the site map? If the major subsurface flow paths have been mapped these should be shown on Fig 1.

-We included a map with all known pathways in Figure 1.

L33-7 This seems to imply that groundwater flow within the matrix only happens during low-rainfall periods, which is not the case. Groundwater flow within the matrix would be continuous, but small relative to the amount of groundwater flowing along conduits
following rainfall events. Also, be careful to be clear on the two separate processes of recharge and groundwater flow, the two seem to be used here almost as if they are the same thing. The relevance of the water quality comments to end this section to the study is not immediately clear, expansion of this paragraph may be helpful.

-We can rephrase this and also expanse the paragraph in a revised version. Additional information can be added from Eiche et al 2016. During dry season, matrix flow is dominant and assures a year-round flow of water. During rainy season, matrix flow is regularly overprinted by piston flow. During wet season, recharge through cracks, dolines etc. dominates which also leads to the fluctuations (EC declines quickly as discharge increases, but values go back to "normal" quickly again when no new rain event occurs).

Pg4

L4 Multiple lines of evidence have been used in this study, but it doesn't seem like multiple methods were actually "compared". This implies that the same types of estimates (i.e. discharge volume) came out of each method and these values were compared, but this is not actually the case.

-We actually used different methods to study land-ocean groundwater nutrient fluxes, this includes SST pictures for the identification, discharge data, precipitation data and nutrient concentrations in groundwater. We don't compare these methods with each other, but we use their results to study the nutrient fluxes.

L15 This sentence needs rewording.

-We can rephrase it.

L17 It seems as though the location of groundwater discharge at the coast was identified prior to this study, the authors should clarify exactly what the new contribution of this SST data is relative to what was known previously.

-The SST data indicates where groundwater discharges into the sea, and at which

locations this groundwater discharge will have a relatively large effect on the coastal ocean (e.g. high discharge, but also longer residence time in the coastal ocean). As an example, discharge at Pantai Ngrumput does not mix a quickly with seawater, as at other sites (e.g. Pantai Sundak). This is one of the reasons why this site has been identified by SST pictures.

L24 The relevance of the Siebert 2014 reference here is not clear. "groundwater uninfluenced" is a rather clunky way of putting it.

-We will rephrase this in a revised version of the manuscript

L28 Figure 3 is referred to prior to Figure 2 (not cited until pg 6)

-We could change the order of both Figures.

Pg5

L1-4 What is the relevance of the pH, and DO data to the manuscript?

-We agree that pH may be left out, as it does not give any relevant information in the manuscript. DO values are interesting as they show that groundwater was oxic. This has a large impact on nitrogen turnover in the aquifer (e.g. nitrification). This information can be included in a revised version of the manuscript.

L5 Delete "In Germany" and correct "isotopy"

-We will do this in a revised version.

L8 Are the nutrients also dissolved? Or ar these total NO3 etc.? If you filtered then aren't these dissolved? And why didn't you estimate fluxes of all nutrients, why only Si and NO3?

-They are dissolved. NO3 was the dominant form of total nitrogen, while very little ammonium and nitrite were observed. Therefore we show only NO3 fluxes in terms of nitrogen. PO4 and DSi fluxes will be shown in a revised manuscript as well (Figure 5).

Section 3.3 The assumptions made in this section do not seem overly simplistic, as mentioned above. Event-scale variation in stable isotopic values may be able to back up/test these assumptions – i.e. depleted signature during heavy rainfall.

-In a revised version of the manuscript we identify 4 events with high discharge, which followed heavy rain events (Figure 2). These events can also be identified based on a lower EC and a lower temperature at Pantai Baron. Some of these events correlate with higher NO3 concentrations while others don't. We explain for these variations in NO3 concentrations in a revised version of a manuscript, by several processes (e.g. recharge from Epikarst, dilution, anthropogenic inputs).

Section 4.1 First paragraph is not results and is mostly repeated from earlier in the manuscript.

-We can change this accordingly. This was also suggested by Reviewer 1.

Pg6

L3-19 The value of this detailed description of precip data to the manuscript is not clear. A more concise treatment may be to calculate the time lags between precip and groundwater flow or discharge at the coast, rather than a full description of each event, which can be seen on Fig 2 anyway.

-We can discuss precipitation events in Ponjong and the discharge at Bribin (e.g. Figure 2) as both stations are located close to each other. The remaining climate stations in the study area will not discussed in detail in a revised version, but nevertheless shown in the appendix.

L22-23 "to which it is bounded" isn't quite right. Avoid the use of the word "shows", as it is not the correct word.

-We will rephrase this.

L28-32 The relevance of this stable isotope analysis to either groundwater recharge,

flow or discharge at the coast is not clear.

-We would remove the stable isotope data from a revised version of the manuscript.

L34-35 Why define DSi? Why not just use Si – NO3 is also dissolved isn't it? Fig 5A+B – just call it Fig 5 (it only has the two parts).

-DSi is a general term for dissolved silicon, which includes all different types of dissolved silicon (anything smaller than 0.45 $\mu$m in this context). We do not see any necessity of changing the term DSi in this context.

L36-37 Avoid using terms relative terms like similar (without specifying what it is similar to) and high (what exactly does "high temporal variability" mean?)

-We will use a different wording in a revised manuscript.

Pg7

L5-15 Can you do some stats to back up your interpreted relationships? Are NO3 and Si correlated with subsurface flow? Or are there lag times between peak NO3 and peak flow rates? Also, what is the value of the Si data in this analysis? It doesn't seem link back to your stated aims.

-DSi is also considered as a nutrient as it is used by diatoms in coastal waters, and therefore show DSi as a nutrient. We follow the suggestion of reviewer 2 and correlate NO3, DSi and PO4 with discharge. We can see a dilution effect for DSi, which indicates a baseflow DSi signal which is diluted during heavy rain events. This is not the case for NO3 (Figure 4) indicating that there are several sources and processes controlling NO3 concentrations in the aquifer. We will discuss these processes in the discussion of a revised version of the manuscript.

Discussion: The discussion contains references to a number of figures, which suggests that these comments should have been made in the results section. The discussion should not present or highlight new information about the data that wasn't already

presented in the Results section.

-We will avoid showing results in the discussion section.

L19 Avoid general terms such as "a major amount" and "A further part".

-We will use a different wording.

L22 A lag time of 4 days between precip and groundwater flow is not self-evident from Fig 2, and why was this not highlighted in the results section? Previously you mentioned the lag times as having been assumed from the results of previous studies.

-Lag times of groundwater travel times have been assumed from previous studies. Here we can explain high discharge events at Bribin-Sindon with heavy rain events which were observed in general about 1 day before. We can include this into a revised version of the manuscript.

L25-27 (and Figure 4) Given rapid infiltration and recharge during rainfall events, why do the stable isotope values not plot on the local meteoric water line?

-This local meteoric water line has been published by Sidauruk (2015). However, the local Meteoric water line of the region is just based on a limited amount of samples and probably not completely reliable yet. This is one reason why we are limited in interpreting the stable isotopic data and why we would therefore rather leave it out of the manuscript.

L30-37 This discussion seems tangential to the current study. You have said in your introduction that rainfall events increase turbidity in the subsurface, and referenced your earlier paper. It doesn't seem like this paper has contributed anything new to our understanding of E. Coli or tourism. The majority of 5.1 seems to have already been covered by Eiche et al 2016.

-While Eiche et al 2016 investigated different types of flow (matrix flow, piston flow) our study investigates nutrient fluxes and its sources. We will try to be more specific

towards the nutrient fluxes in a revised version of the manuscript.

Pg8

L4-10 This paragraph discusses correlation between data sets, but no correlation statistics were reported in the results section. What does it mean to have both positive and negative correlation? Does this mean there actually isn't a correlation if you look at the full data set?

-In a revised version we will show correlation statistics, as mentioned before.

L11-19 The relevance of the Si data and interpretation to the stated aims of the paper are not clear. You say here that the Si concentrations are diluted during low flow events, does this then support your interpretation that NO3 stores must be released from the unsaturated zone during floods? On Fig 5, during the recession period where you actually have good temporal resolution of data, Si increases while NO3 decreases, what does this mean in terms of process? You write "a further DSi source" do you mean further spatially, or an additional source? The relevance of the comments on colloidal transport to the current study is not clear.

-The primary source of DSi might be volcanic deposits e.g. towards the north or underlying the carbonates. This setting leads to high DSi concentrations, in comparison to many other non volcanic karstic regions in which groundwater is often characterized by low DSi concentrations. This is an interesting point, because it shows that karstic regions in tropical volcanic islands also transport high amounts of DSi. DSi may be transported into groundwater from matrix flow, or recharge from the volcanic hinterland. NO3 input is in turn controlled by many factors which may vary over temporal scales. Nitrogen from sewage is more or less constant during the year but an increase in flushing into the aquifer and thus into solution does happen just after rain events. In between, this nitrogen source can seep into the underground via matrix flow or is accumulated. The other sources like fertilizer are just brought in during certain times. If during that time, rain events take place, even small ones, than Nitrate is washed into

the subsurface water sources. In a revised version of the manuscript we would discuss the possible sources of nutrients in depth.

L20-35 Delete "from these fertilizers" at L30.

-We will delete this from the manuscript.

What is "temporal exhaust" on L 35?

-It means that nitrate is exhausted in the soil after flood periods, while it accumulates again in the surface of the soil during dry periods.

Pg9 L1-3 This seems more like an introductory statement. Correlation is mentioned again, but stats not reported.

-We will omit the word correlation in this context

Section 5.3 The treatment of uncertainties is inadequate given the assumptions in the analysis (see comments above). This section identifies some sources of error, but does not actually report any quantified uncertainties.

-We agree with the reviewers that we have to deal with this uncertainty in a much better way. We will quantify the error by show minimum and maximum nutrient fluxes.

L9-10 What do you mean by "A general connection was deduced from temporal variability of hydrochemistry"?

-We will change this statement and mention the correlation matrix (Spearman's rank) (Table 3) which shows a similar change in nitrate concentration with time.

L18-21 The conclusion begins by acknowledging that a vast area of hinterland may contribute to nutrient discharge at the coast, so it is not clear how does the spatially sparse data set (on groundwater sampling location) can provide a robust estimate of nutrient fluxes.

-The locations where the actual water can be sampled on a regular basis are sparse.

Influx of nutrients often occurs through sinkholes which cannot be measured. Furthermore, many underground rivers are not easily reached for sampling which also limits the available sites. Also it should be emphasized that staff and money is limited in such a study so not everything can be sampled at high resolution. To our knowledge a study on land-ocean groundwater nutrient fluxes in a tropical karstic region has not yet been published with such an extensive dataset, especially not in Indonesia, where logistics can be complicated. Furthermore our datasets suggest similar controlling mechanisms at different sampling sites, for example large variations nitrate concentrations from up to 300 $\mu$mol/L down to 0 $\mu$mol/L. This can all be explained by a combination of high anthropogenic activity combined with a high temporal variability in precipitation and discharge e.g. between wet and dry season.

L28 What is "highly variable"?

-We can rephrase this. We mean that groundwater nutrient fluxes are variable over temporal scales.

L32-35 (and L1-2 Pg 10) This seems to be suggesting a better sampling design for the current study, to capture temporal variability by measuring at a higher temporal resolution.

-Our study shows that groundwater nutrient fluxes are temporally highly variable in a tropical karstic environment. We suggest best possible sampling times for any further research in these areas. We do not question our sampling design, which was done the best way we could get.

References:

Eiche 2016 title is incomplete.

-We will change this in a revised version of the manuscript.

Fig 1. There is a light blue colour in the mid-left of the map that doesn't seem to be explained in the legend.

-We will show an updated legend in a revised version of the manuscript.

Fig 2 Why are the discharge data so patchy?

-Such a high temporal resolution of discharge in an underground river (every 10 min) can rarely be found so in our view it is not patchy. The dam is used to produce electricity so that water can be provided to the surrounding villages. To assure a long-term success of this prototype it regularly has to shut down in order to carry out important maintenance work. During that time, discharge measurements are not possible. Furthermore, electricity failures (which are common in the area) also have occurred during the time period that has been shown. This explains some of the gaps. Everybody who has worked in this or a similar area is probably aware of the difficulty to produce high resolution data without any gaps.

Fig 4 X-type symbols are too similar.

-In a revised version we won't show stable isotope data.

Fig 6 Is the discharge shown on the day of the flux estimate, or the day used to calculate the flux estimate? The discharge on the day NO3 concs were measured, or the discharge 4-14 days prior? Why use a bar chart instead of a time series?

-In a revised version of the manuscript we show a range of nutrient fluxes for the data of sampling at Pantai Baron. The nutrient flux estimates are based on discharge data which was measured at Bribin Sindon 12 to 16 days prior to sampling during non flood and 2 to 6 days prior to sampling during flood events.

Table 1 Use delta, not d for isotopes. NO2, NH4 and PO4 data are not discussed in any detail in the manuscript. Nutrients and NO3 are not the same thing. Is it a paper on all of these nutrients, or just on nitrate fluxes?

-In a revised version we show NO3, DSi and PO4 fluxes (Figure 5). NO2 and NH4 are not shown in detail as the dominant part of nitrogen occurs in the form of nitrate. However, we can discuss in more detail that NO3 is the dominant form because groundwater is oxic promoting nitrification.

Table 2 Caption says average discharge rates, but table reports avg (presumably) as well as min and max, average and standard deviation would be more concise. Table should indicate which are dry season and wet season samples (=4 or 14 days prior to concentration measurement). Some uncertainty on the flux estimates should be provided. Why are fluxes of NH4 and PO4 not provided? Measured concentrations used to calculate fluxes should also be reported in this table.

-We include a minimum and maximum discharge rate and flux for these datasets (Table 4). Flooding and non-flooding events are marked by grey lines in the table. In a revised version of the manuscript we would include this table into the supplementary material.

––––––––––––––––––––––

[Figure]

Fig. 1 map content:

**Legend**

| | |
|---|---|
| | Limestone |
| | Alluvium |
| | Volcanic breccia, lava, tuff |
| | Marl, Tuff |
| | Tuff, Sandstone |
| | Sandstone Claystone |
| | Marl, Limestone |
| ◊ | Climate station |
| ● | Coastal spring |
| ○ | Subsurface river |
| ■ | Subsurface river dam |
| ····· | NoFlowPaths |
| -·-→ | FlowPaths |

**Fig. 1.** All proven land-ocean groundwater connections (black lines) and negotiated connections (red lines) in the karstic region of Gunung Kidul. For respective discharge rates and names and types o

**Fig. 2.** TIR image of the coastal ocean showing two major sites of groundwater discharge and related electrical conductivity values which were measured in the coastal water.

[Figure]

**Fig. 3.** Discharge at the subsurface river dam (grey) and precipitation data (blue) from the upstream located climate station Ponjong, nitrate concentrations at the coastal spring Pantai Baron (red dots). The

**Fig. 4.** Scatterplot of DSi, NO3, and PO4 in relation to the minimum discharge at Pantai Baron.

[Figure]

**Fig. 5.** Range of land-ocean groundwater nutrient fluxes estimated based on groundwater discharge rates from a subsurface river dam and nutrient concentrations sampled at Pantai Baron. The upper white part of

| Site | Map ID | Type | Discharge (m³/s) dry season | Discharge (m³/s) wet season | Comment | Reference |
|---|---|---|---|---|---|---|
| P. Baron | a | Coastal spring | 4-8.2 | | | 1 |
| P. Ngrumput | b | Coastal spring | 0.05-0.06 | 0.03 | | 2 |
| P. Slili | c | Coastal spring | 0.05 | | No connection to Bribin-Sindon | 1 |
| P. Sundak | d | Coastal spring | 0.2 | | No connection to Bribin-Sindon | 1 |
| Pok Tunggal | e | Coastal spring | | | No connection to Bribin-Sindon | |
| P. Ngobaran | f | Coastal spring | | | No connection to Bribin-Sindon | |
| Buhputih | g | Subsurface river | 0.02 | | Flows to Baron | |
| Bribin-Sindon | h | dam | >1 | <12 | Flows to Baron | |
| Gua Bribin | h | Subsurface river | 1-1.3 | 4-8 | Gua Bribin is 2 km upstream of Bribin-Sindon | 1,3 |
| Seropan | i | Subsurface river | 0.4-0.5 | 0.5 to <3, extreme >10 | Flows via Ngreneng to Baron | 3,4 |
| Grubug | j | Subsurface river | 0.7-1 | 2 | 100% flows to Baron 25% of discharge of Baron | 1,3 |
| Gua Ngreneng | k | Subsurface river | <0.1 | 0.2 | | 1,3 |
| Luweng Jomblangan | l | Subsurface river | | | Flows to Bribin-Sindon | 1,3 |
| Gilap | m | Subsurface river | 0.003 | | Flows to Bribin-Sindon | 1,3 |

* 1 = MacDonalds&Partners 1984; 2 = own measurements 2016; 3 = own measurements 2000/2001; 4 = own measurements 2008-2010

**Fig. 6.** Table 1: All known discharge rates measured at subsurface rivers in the hinterland and coastal springs are shown in this table. The site where the measurement was taken (Flow ID) is shown in Figure 1.

| Event | Date | Season | DO (%) | EC (µS/cm) | Temp (°C) | NO$_3$ (µmol/L) | NO$_2$ (µmol/L) | NH$_4$ (µmol/L) | DSi (µmol/L) | PO$_4$ (µmol/L) |
|---|---|---|---|---|---|---|---|---|---|---|
| Pantai Baron-1 | 14-Nov-2015 | Dry | 83 | 557 | 27.9 | 170 | 0.0 | | 408 | 0.1 |
| Pantai Baron-2 | 19-Apr-2016 | Wet | 85 | 429 | 27.6 | 114 | 0.3 | 1.3 | 335 | 0.1 |
| Pantai Baron-3 | 10-May-2016 | Dry | 88 | 521 | 28.0 | 90 | 0.2 | 0.6 | 458 | 0.1 |
| Pantai Baron-4 | 24-May-2016 | Dry | 82 | 541 | 28.1 | 21 | 0.4 | 3.6 | 471 | 0.1 |
| Pantai Baron-5 | 8-Jun-2016 | Dry | 82 | 525 | 28.0 | 78 | 0.3 | 2.4 | 436 | 0.1 |
| Pantai Baron-6 | 21-Jun-2016 | Dry | 77 | 384 | 27.2 | 27 | 0.2 | 1.2 | 297 | 0.2 |
| Pantai Baron-7 | 20-Aug-2016 | Dry | | 640 | 27.8 | 271 | | | | |
| Pantai Baron-8 | 23-Sep-2016 | Dry | | 820 | 28.4 | 115 | | | | |
| Pantai Baron-9 | 30-Nov-2016 | Wet | | 260 | 23.0 | 52 | | | | |
| Pantai Baron-10 | 13-Dec-2016 | Wet | 92 | 429 | 27.6 | 140 | 0.4 | | 312 | 0.9 |
| Gua Pindul-1 | 21-Apr-2016 | Wet | 92 | 533 | 28.6 | 94 | 0.6 | 1.9 | 320 | 0.1 |
| Gua Pindul-2 | 11-May-2016 | Dry | 88 | 540 | 28.5 | 48 | 0.3 | 1.9 | 389 | 0.1 |
| Gua Pindul-3 | 24-May-2016 | Dry | 87 | 567 | 27.9 | 72 | 0.3 | 1.4 | 413 | 0.1 |
| Gua Pindul-4 | 8-Jun-2016 | Dry | 86 | 558 | 28.4 | 33 | 0.3 | 2.1 | 410 | 0.1 |
| Gua Pindul-5 | 21-Jun-2016 | Dry | 86 | 413 | 27.3 | 34 | 0.4 | 3.3 | 303 | 0.1 |
| Gua Pindul-6 | 20-Aug-2016 | Dry | | 520 | 29.6 | 302 | | | | |
| Gua Pindul-7 | 23-Sep-2016 | Dry | | 560 | 29.0 | 0 | | | | |
| Gua Pindul-8 | 30-Nov-2016 | Wet | | 270 | 25.6 | 0 | | | | |
| Gua Pindul-9 | 12-Dec-2016 | Wet | 99 | 494 | 26.1 | 84 | 1.1 | | 316 | 0.1 |
| Gunung Kendil-1 | 17-Apr-2016 | Wet | 93 | 567 | 27.8 | 123 | 0.5 | 4.1 | 495 | 0.1 |
| Gunung Kendil-2 | 11-May-2016 | Dry | 84 | 522 | 27.4 | 91 | 0.0 | 0.0 | 462 | 0.1 |
| Gunung Kendil-3 | 24-May-2016 | Dry | 82 | 524 | 27.2 | 53 | 0.2 | 1.4 | 497 | 0.1 |
| Gunung Kendil-4 | 8-Jun-2016 | Dry | 73 | 527 | 27.0 | 66 | 0.2 | 0.2 | 474 | 0.1 |
| Gunung Kendil-5 | 21-Jun-2016 | Dry | 82 | 532 | 27.1 | 64 | 0.0 | 0.0 | 450 | 0.1 |
| Kali Suci-1 | 21-Apr-2016 | Wet | 104 | 426 | 29.1 | 58 | 0.2 | 1.1 | 308 | 0.1 |
| Kali Suci-2 | 10-May-2016 | Dry | 102 | 431 | 28.1 | 107 | 0.3 | 1.7 | 362 | 0.1 |
| Kali Suci-3 | 24-May-2016 | Dry | 101 | 485 | 27.5 | 66 | 0.2 | 0.9 | 408 | 0.1 |
| Kali Suci-4 | 8-Jun-2016 | Dry | 104 | 490 | 27.5 | 133 | 0.2 | 0.6 | 382 | 0.1 |
| Kali Suci-5 | 21-Jun-2016 | Dry | 102 | 417 | 26.7 | 69 | 0.1 | 0.3 | 307 | 0.1 |
| Kali Suci-6 | 20-Aug-2016 | Dry | | 520 | 28.9 | 230 | | | | |
| Kali Suci-7 | 23-Sep-2016 | Dry | | 490 | 29.2 | 0 | | | | |
| Kali Suci-8 | 30-Nov-2016 | Wet | | 200 | 26.7 | 38 | | | | |
| Pantai Ngrumput-1 | 16-Nov-2015 | Dry | 75 | 8380 | 27.8 | 132 | 0.0 | | 372 | 0.5 |
| Pantai Ngrumput-2 | 19-Apr-2016 | Wet | 72 | 6300 | 28.4 | 30 | 0.1 | 7.9 | 350 | 0.1 |
| Pantai Ngrumput-3 | 10-May-2016 | Dry | 83 | 9530 | 28.8 | 17 | 0.2 | 7.4 | 368 | 0.1 |
| Pantai Ngrumput-4 | 21-Jun-2016 | Dry | 72 | 9450 | 28.4 | 7 | 0.4 | 5.1 | 391 | 0.1 |
| Pantai Ngrumput-5 | 20-Aug-2016 | Dry | | 7520 | 28.2 | | | | | |
| Pantai Ngrumput-6 | 23-Sep-2016 | Dry | | 7510 | 28.5 | | | | | |
| Pantai Ngrumput-7 | 30-Nov-2016 | Wet | | 8080 | 27.2 | | | | | |
| Pantai Ngrumput-8 | 13-Dec-2016 | Wet | 67 | 5950 | 28.2 | 145 | 0.0 | | 302 | 1.0 |

= base flow at Pantai Baron
= high discharge event at Pantai Baron

**Fig. 7.** Table 2: The hydrochemistry of the springs which are located in the hinterland and at the coast. High discharge events at Pantai Baron are marked by the grey shaded areas.

|  | P. Ngrumput | Gunung Kendil | Kali Suci | Goa Pindul |
|---|---|---|---|---|
| P. Baron | 0.90 | 1.00 | 0.12 | 0.44 |
| P. Ngrumput |  | 1.00 | -0.50 | 0.80 |
| Gunung Kendil |  |  | -0.10 | 0.30 |
| Kali Suci |  |  |  | 0.54 |

**Fig. 8.** Table 3: Correlation matrix (Spearman's rank) of temporal NO3 concentration variations of the different springs which were sampled in Gunung Kidul.

| Date | P. Baron min discharge (m³/sec) | P. Baron max discharge (m³/sec) | NO3 (mol/m³) | NO3 min flux (10^6 mol/day) | NO3 max flux (10^6 mol/day) | DSi (mol/m³) | DSi min flux (10^7 mol/day) | DSi max flux (10^7 mol/day) | PO4 (mol/m³) | PO4 min flux (10^4 mol/day) | PO4 max flux (10^4 mol/day) |
|---|---|---|---|---|---|---|---|---|---|---|---|
| Nov-15 | 4 | 5 | 170 | 63 | 74 | 408 | 15 | 18 | 0 | 5 | 6 |
| Apr-16 | 10 | 13 | 114 | 94 | 129 | 335 | 28 | 38 | 0 | 12 | 17 |
| May-16 | 1 | 8 | 90 | 11 | 65 | 458 | 6 | 33 | 0 | 2 | 10 |
| May-16 | 4 | 6 | 21 | 7 | 11 | 471 | 15 | 23 | 0 | 5 | 7 |
| Jun-16 | 2 | 4 | 78 | 16 | 30 | 436 | 9 | 17 | 0 | 2 | 4 |
| Jun-16 | 3 | 8 | 27 | 6 | 18 | 297 | 7 | 20 | 0 | 4 | 13 |
| Aug-16 | 5 | 5 | 271 | 111 | 123 | | | | | | |
| Sep-16 | 5 | 5 | 115 | 46 | 53 | | | | | | |
| Nov-16 | 5 | 13 | 52 | 22 | 59 | | | | | | |
| Dec-16 | 11 | 20 | 140 | 133 | 245 | 312 | 30 | 55 | 1 | 85 | 157 |

☐ = base flow at Pantai Baron

▨ = high discharge event at Pantai Baron

**Fig. 9.** Table 4: Range of groundwater discharge rates, NO3 fluxes, DSi fluxes and PO4 fluxes at Pantai Baron. Flooding events are marked by the grey line.